# Adaptive expansion of ERVK solo-LTRs is associated with Passeriformes speciation events

Guangji Chen [1,2,3], Dan Yu[2,4], Yu Yang[5], Xiang Li [6], Xiaojing Wang[6], Danyang Sun[2,4], Yanlin Lu[2,4], Rongqin Ke [5], Guojie Zhang [2,7], Jie Cui [8,9,10,11] ✉ & Shaohong Feng [2,7,12] ✉

Endogenous retroviruses (ERVs) are ancient retroviral remnants integrated in host genomes, and commonly deleted through unequal homologous recombination, leaving solitary long terminal repeats (solo-LTRs). This study, analysing the genomes of 362 bird species and their reptilian and mammalian outgroups, reveals an unusually higher level of solo-LTRs formation in birds, indicating evolutionary forces might have purged ERVs during evolution. Strikingly in the order Passeriformes, and especially the parvorder Passerida, endogenous retrovirus K (ERVK) solo-LTRs showed bursts of formation and recurrent accumulations coinciding with speciation events over past 22 million years. Moreover, our results indicate that the ongoing expansion of ERVK solo-LTRs in these bird species, marked by high transcriptional activity of ERVK retroviral genes in reproductive organs, caused variation of solo-LTRs between individual zebra finches. We experimentally demonstrated that *cis*-regulatory activity of recently evolved ERVK solo-LTRs may significantly increase the expression level of *ITGA2* in the brain of zebra finches compared to chickens. These findings suggest that ERVK solo-LTRs expansion may introduce novel genomic sequences acting as *cis*-regulatory elements and contribute to adaptive evolution. Overall, our results underscore that the residual sequences of ancient retroviruses could influence the adaptive diversification of species by regulating host gene expression.

Endogenous retroviruses (ERVs) are viral sequences that have been integrated into a host's genome as residue from past retroviral infections of germ cells[1]. ERVs genome consists of three main genes: *gag* and *pol* encoding the necessary proteins for replication, and *env* encoding surface proteins that help the virus enter host cells, and paired long terminal repeats (LTRs), which are located at the flanks of the ERV[2]. The high similarity among paired LTRs leads to unequal homologous recombination, which results in the deletion of the internal region, as well as the formation of residual solitary LTRs (solo-LTRs) and the functional decay of the viral DNAs[3]. Studies on plants show that LTR retrotransposons can constitute an enormous proportion of the total genome (e.g., 75% of the genome of maize)[4], and suggest that solo-LTRs formation may drive the evolution of genome size[5]. LTR retrotransposons are also prevalent in the genomes of amniotes; for instance, they constitute ~8% of the human genome[6].

In birds, which represent a highly diverse group of amniotes, LTR retrotransposons constitute a notably low proportion of the total genome, at only 0.2–5%[7]. It has been hypothesized that adaptations to meet the metabolic demands of a flight lifestyle account for the relatively small sizes of bird genomes (0.9 ~ 1.3 G)[8,9], which comprise fewer transposable elements (TEs) and experience less frequent paleoviral infiltrations[8,10,11]. However, as studies have shown that the shrinkage of

bird genomes may have occurred long before the evolution of flight[12], other mechanisms may be responsible for reductions in the genome sizes of birds[13]. One alternative hypothesis is that the loss of DNA by deletions that countered DNA acquisition via transposon expansion may have had a major role in maintaining the small and stable sizes of bird genomes[11]. Specifically, the formation of solo-LTRs may have led to the removal of the DNA that had accumulated by the amplification ERVs, offering a mechanism to balance the reduction of DNA by the acquisition of new DNA. Hence, the ratios of deleted to acquired DNA can serve as indicators of the potential strength of selection pressures on the ERV amplification, offering a way to understand the distribution pattern of ERVs in modern bird genomes.

Although most ERVs have lost their functions as a result of accumulated mutations or internal recombination, their occurrences within host species can be evidenced functionally. Several studies have shown that ERVs can manifest their biological functions through viral proteins[14,15]. For instance, in humans, the widely studied *syncytin1* gene, derived from the human endogenous retrovirus W (HERV-W) *env*, is directly involved in the fusion and differentiation of the trophoblast, and plays essential roles in placental development and syncytial formation[16]. Moreover, human endogenous retrovirus K (HERV-K), the most highly active human endogenous retrovirus family[17], is widely expressed in many human tissues[18]. The activity of HERV-K has been found to be associated with various neurodegenerative diseases, including amyotrophic lateral sclerosis (ALS)[19], schizophrenia[20], as well as cancers[21]. Additionally, some studies report that the LTRs of ERVs may contain primary promoters and regulatory elements for provirus expression[22], while other studies have employed these regulatory elements to regulate expressions of host genes[23,24]. For example, HERV LTRs have been co-opted as enhancers of the interferon-stimulated *AIM2* gene in humans to activate inflammatory responses via interferon-γ-inducible regulatory networks[25]. Notably, the regulatory functions of LTRs could be tissue-specific. For instance, the LTRs of human endogenous retrovirus P (HERV-P) were found to function as a tissue-specific promoter of the *NAIP* gene in human testes and prostate[23], which encodes for the neuronal apoptosis inhibitory protein and plays an important role in testicular function. More generally, studies have proposed that the evolution of new traits can be achieved through the specific regulation of gene expression via variations in non-coding sequences[26].

Given that ERVs can introduce novel genetic materials and biological functions in their hosts, the acquisition of specific ERVs by some host species may have been adaptive and contributed to the evolution of novel traits. In the present study, we aim to understand the evolution of ERVs, especially the underestimated solo-LTRs, in birds. Specifically, focusing on the genomes of 362 avian species – generated from the Bird 10,000 Genomes (B10K) Project – we investigate the effects of solo-LTRs in shaping the genomic architecture of their hosts and in introducing regulatory elements over evolutionary time. In comparison with reptilian and mammalian species, we find that birds have higher frequencies of solo-LTRs formation, which is indicative of their higher efficiency in purging endogenous viral elements (EVEs) from their genomes. By integrating data on transcription expression and population re-sequencing in the zebra finches (*Taeniopygia guttata*), we demonstrate the possibility of the continuous accumulation of endogenous retrovirus K (ERVK) solo-LTRs throughout the speciation of Passeriformes, especially for species within the parvorder Passerida. In addition, we detect 20 genes in zebra finches that harbor lineage-specific ERVK solo-LTRs and are differentially expressed in comparison with their orthologs in chickens. We then functionally demonstrate that one of these genes, *ITGA2*, is differentially expressed in the brain regions of zebra finches using fluorescence in situ hybridization (FISH). Additionally, using dual-luciferase reporter assay experiments, we validate the *cis*-regulatory activity of 405-bp ERVK solo-LTRs, which are located upstream of *ITGA2* and thus may be involved in the regulation of the host gene.

## Results

### Higher frequencies of solo-LTRs formation in avian genomes compared with mammals and reptiles

To investigate the compositions of solo-LTRs among different classes of amniotes, we identified solo-LTRs from the genomes of 405 representative amniote species, including 362 birds, 23 reptiles, and 20 mammals (Fig. 1a, Supplementary Fig. 1 and Supplementary Data 1) and constructed the databases of the non-redundant LTR sequences in amniotes. While a significant positive correlation between genome size and the number of solo-LTRs was observed in the genomes of reptile and mammal species, the genomes of bird species did not show this pattern (Fig. 1b and Supplementary Fig. 2). The proportions of solo-LTRs relative to genome sizes in birds were significantly lower than those for mammals (Welch's one-sided t-test, p-value = 0.0001; inner circle in Fig. 1a), but not significantly lower than those for reptiles (Welch's two-sided t-test, p-value = 0.5125; inner circle in Fig. 1a). These differences may have been due to the relatively lower proportion of ERVs occurring within bird genomes. Next, we used the ratio of solo-LTRs to total LTRs length across the genome as the frequency of solo-LTRs formation, indicating the efficiency with which a host species had removed EVEs from its genome. We found that birds had significantly higher frequencies of solo-LTRs formation in comparison with reptiles and mammals (Welch's one-sided t-test, p-value = 0.0003 and 0.0019, respectively; outer circle in Fig. 1a). Our results suggest that these higher frequencies of solo-LTRs formation appeared as early as in Archelosauria, the common ancestor of turtles and archosaurs (as well as birds and crocodilians), as the frequencies were also relatively high in Crocodilia and Testudines (Welch's one-sided t-test between Archelosauria with other reptiles, p-value = 0.0015; outer circle in Fig. 1a). We observed that the high frequencies of solo-LTRs formation have persisted across all bird lineages; even in the recent lineages of Passeriformes, the ratio was maintained at 97.6% on average, indicating a selection force might have functioned to maintain a low proportion of EVEs in the genome. These results suggest that birds could more effectively purge EVEs from their genomes. This self-protection mechanism, together with the lower frequency of paleoviral infiltration suggested by the previous study[10], contribute to the current landscape of low EVEs in bird genomes. Studies have found that the genomes of the two Neoaves groups Piciformes and Bucerotiformes contain substantially higher proportions of transposable elements (TEs) than the genomes of other bird lineages[11,27]. Consistent with this, we observed notably higher proportions of solo-LTRs (5.59% and 2.10%, respectively, the inner circle in Fig. 1a) in species of Piciformes and Bucerotiformes. However, they still have the high frequencies of solo-LTRs formation as other avian species (93.4% and 97.2%, respectively, the outer circle in Fig. 1a). This result further suggested that the higher efficiency of ERV removal through the formation of solo-LTR tends to occur in bird species with higher proportions of ERV insertion.

### The diversification of Passeriformes was accompanied by the accumulation of ERVK solo-LTRs

We further investigated the evolutionary pattern for each ERV category across bird species. Unlike other ERV solo-LTRs commonly observed in diverse proportions among lineages, ERVK solo-LTRs showed a strikingly different pattern, with the continuous expansion in the Passeriformes group and an accumulation over evolutionary time that has accompanied species diversification in this group, especially in the crown group Passerida (Fig. 2 and Supplementary Fig. 3). We observed that the more recent nodes of Passeriformes have accumulated more ERVK solo-LTRs than the ancestral nodes, with the highest accumulated level of ERVK solo-LTRs appearing in the most recently evolving lineage, Passerida (Fig. 2a). Examining correlations between

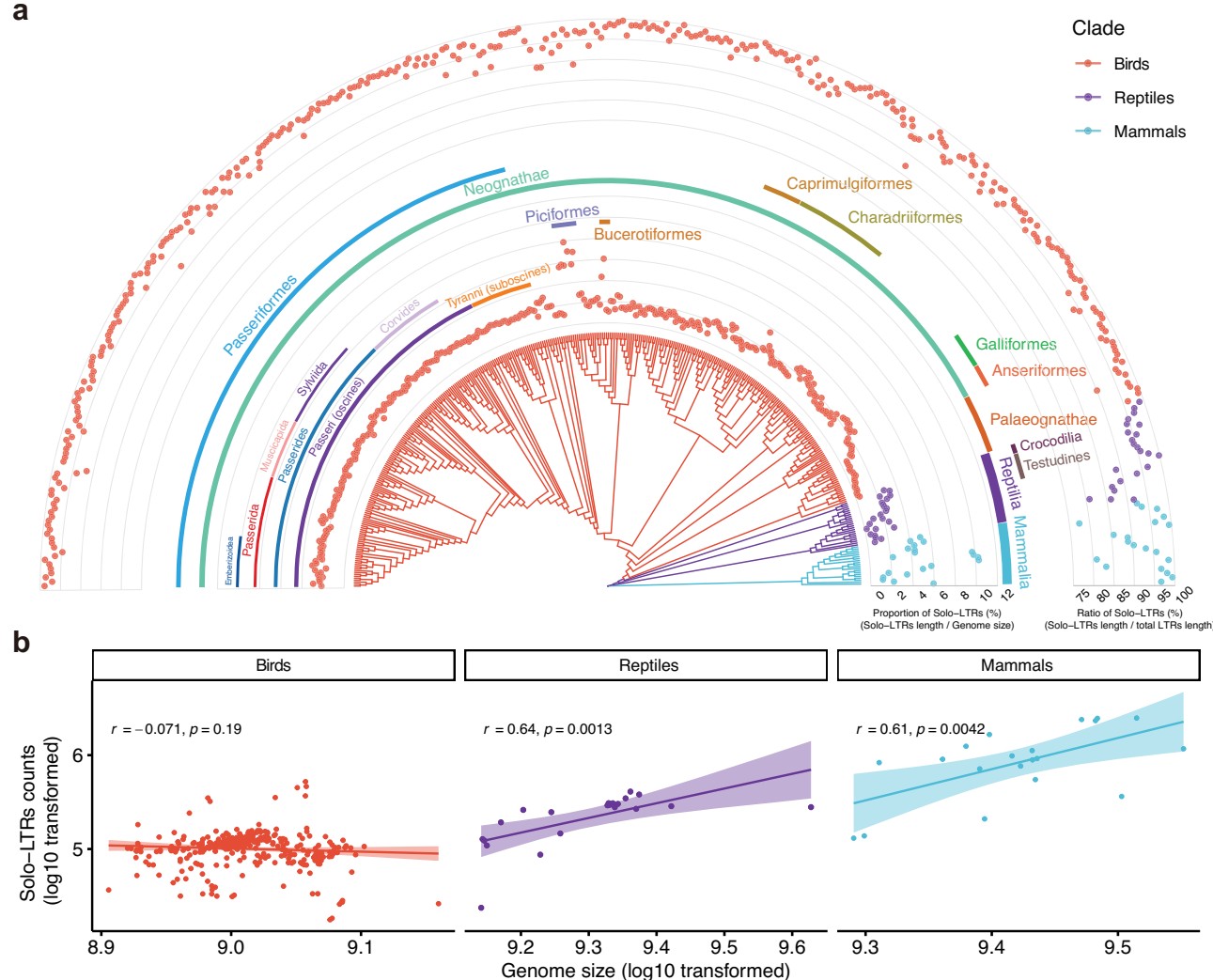

**Fig. 1 | Solo-LTRs in bird genomes display different patterns from those of mammals and reptiles. a** Phylogenetic tree with the proportion of solo-LTRs (the inner circle) and the ratio of solo-LTR (outer circle) in birds ($n = 362$), reptiles ($n = 23$), and mammals ($n = 20$). The inner circle represents the proportion of solo-LTRs relative to the genome size. The outer circle represents the ratio of solo-LTRs to total LTRs length, indicating the frequency of solo-LTRs formation. Taxonomic information follows classifications in Howard and Moore[83]. **b** Solo-LTRs counts were positively correlated with genome size in mammals ($n = 20$) and reptiles ($n = 22$), but not in birds ($n = 345$). Bird and reptile species with potentially problematic assemblies (genome size < 800 Mb or scaffold N50 < 20 kb) were filtered to reduce the bias of assembly quality. Dots correspond to individual species, with red dots indicating bird species, purple dots indicating reptile species, and blue dots indicating mammal species. Correlation analysis was performed by using Pearson correlation at 95% confidence interval, and colored regions indicate the 95% confidence interval for each regression line. Source data are provided as a Source Data file.

the accumulation of ERVK solo-LTRs and speciation events, we found significantly positive correlations at the level of the order Passeriformes as well as the level of the parvorder Passerida (Fig. 2b). This pattern suggested that the speciation process of Passeriformes was accompanied by an accumulation of ERVK solo-LTRs. Moreover, the rate at which ERVK solo-LTRs accumulated had increased during the diversification of Passerida (Fig. 2c). However, other ERV solo-LTRs did not show this ever-accelerating accumulation pattern in Passerida (Supplementary Fig. 4 and Supplementary Data 2). To avoid any potential sampling bias, we used the phylogenetic tree of the 10,135 bird species to re-estimate the correlation between the proportion of ERVK solo-LTRs and the number of species in each family. The accelerating rate of ERVK solo-LTRs accumulation in Passerida could still be observed with broader sampling (Pearson's correlation under 95% confidence interval, $p$-value = $1.84 \times 10^{-5}$; Supplementary Fig. 5 and Supplementary Data 3). Moreover, we also found that the number of annotated ERVK retroviral genes showed a positive correlation with

the proportion of ERVK solo-LTRs in Passerida (Supplementary Fig. 6); this was consistent with the fact that the ERVK provirus has contributed to the accumulation of ERVK solo-LTRs.

### Concurrent accumulation of the ERVK solo-LTRs in the zebra finch

To investigate the invasion of the ERVK provirus during the evolution of Passerida (originated 22.4 MYA), we traced the insertion history of ERVK solo-LTRs based on a whole-genome alignment. We found that on average, approximately 45.72% of ERVK solo-LTRs were species-specific in the suborder Passeri (Fig. 3a). We also found that ERVK solo-LTRs shared between any two Passeri species under the parvorder Passerida made up a large proportion (an average of 76.69%) of all ERVK solo-LTRs in their genomes. However, only a few shared ERVK solo-LTRs which could be traced to the common ancestral nodes of Passeri, Passerides, and Passerida (on average of 8.46%, 0.79%, and 1.97%, respectively). This suggested that the accumulation of ERVK

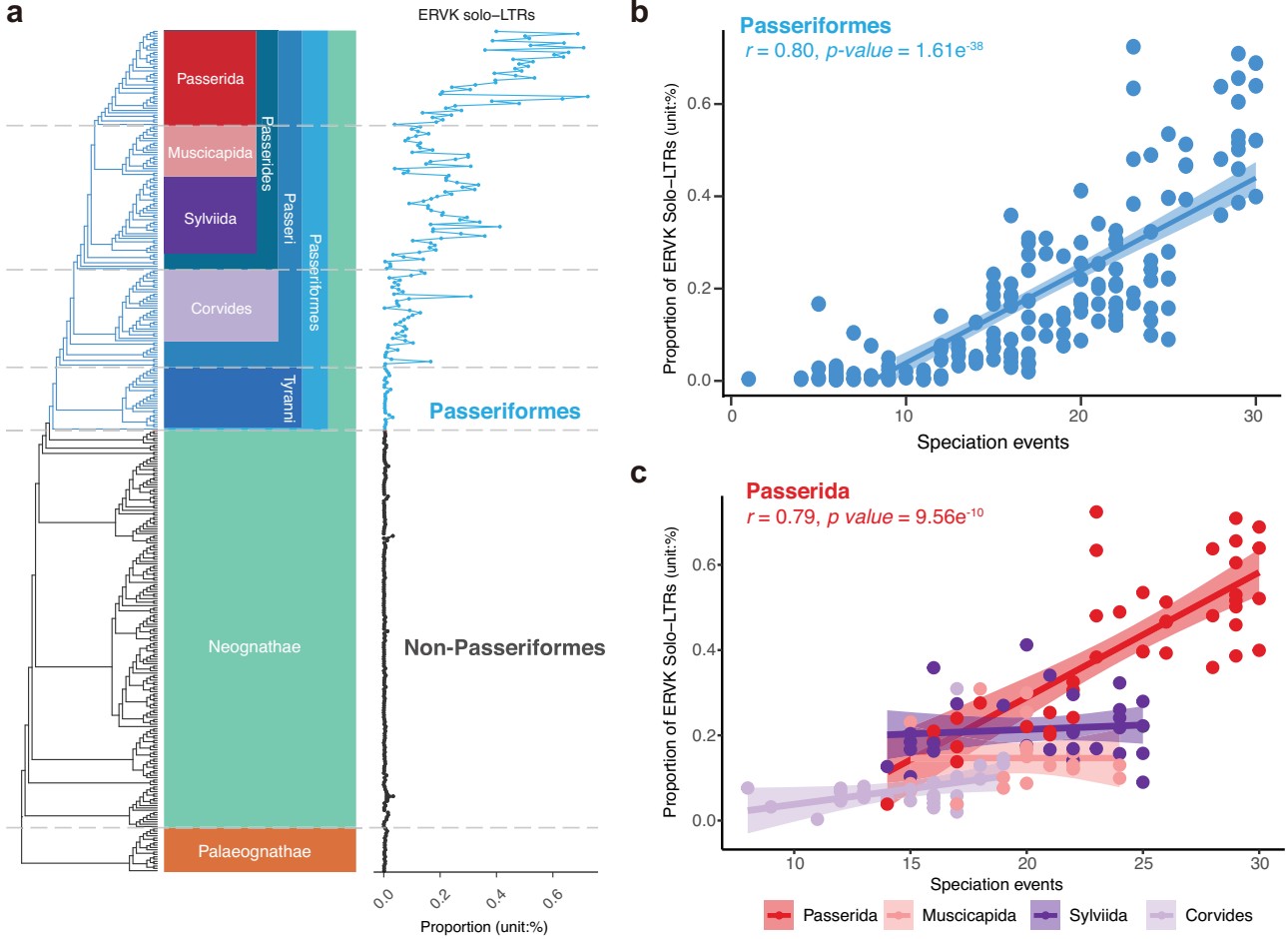

**Fig. 2 | ERVK solo-LTRs accumulate during speciation events in Passeriformes.** **a** Phylogenetic tree illustrating the proportion of ERVK solo-LTRs in 362 bird species. Branches correspond to different species; from top to bottom, the branches above the five gray dashed lines correspond to species in Passerida, Passerides, Passeri, Passeriformes, and Neognathae, respectively. Based on the B10K family-level bird phylogeny. **b** ERVK solo-LTRs positively correlated with the number of speciation events in Passeriformes. **c** The positive correlation can also be observed in the parvorder Passerida. Each dot represents a species and each color of dots represents a clade of birds ($n$ = 169, 41, 20, 33, and 30 for Passeriformes, Passerida, Muscicapida, Sylviida and Corvides, respectively). Speciation events were measured as the number of nodes along a path from the Passeriformes ancestor node to the tips of each species, based on B10K family-level bird phylogeny. Colored regions indicate the 95% confidence interval for each regression line in (**b**, **c**). Correlation analysis was performed by using Pearson correlation at 95% confidence interval. Source data are provided as a Source Data file.

solo-LTRs in Passerida did not occur in the most recent common ancestor (MRCA) of these three lineages, but experienced multiple recent bursts in a lineage-specific way. Throughout the diversification of Passeriformes, it was likely that the ERVK provirus was continually integrated into bird genomes and subsequently eliminated via the formation of solo-LTRs, leading to the accumulated residual of these novel ERVK solo-LTRs.

To assess the activity of the ERVK provirus in modern bird species, we investigated the expression level of ERVK retroviral genes using transcriptome data from seven tissues of the zebra finch (*Taeniopygia guttata*), a representative species of Passerida. We found that 120 ERVK retroviral genes had transcriptional activity in at least one tissue, 75 of which were expressed at significantly higher levels in the ovary, testis, or primordial germ cells (PGC) than in other organs (Fig. 3b). We also found that in the reproductive organs, ERVK retroviral genes had significantly higher levels of expression than other retroviral genes (Welch's one-sided t-test, $p$-value = $6.5 \times 10^{-12}$; Supplementary Fig. 7). Such higher transcriptional activity of the ERVK provirus in the reproductive organs, particularly in the primordial germ cells (Fig. 3b), was the prerequisite for the continual integration of the ERVK provirus in the genome of the zebra finch.

If the expansion of ERVK were an ongoing process, we would expect to observe a signal of recent population diversity in ERVK solo-LTRs. To verify this, we further investigated the polymorphic status of ERVK solo-LTRs in populations of zebra finch by assessing insertion variations among individuals. Using whole genome re-sequencing data from 19 zebra finch individuals[28], we found that 655 (2.68%) ERVK solo-LTRs were polymorphic among individuals (Fig. 3c), supporting the concurrent expansion scenario for some ERVK solo-LTRs in contemporary populations of this species.

### ERVK solo-LTRs are retained as regulatory elements in bird genomes

Previous studies have shown that LTRs may function as *cis*-elements and regulate gene expression in host genomes[23–25]. To gain insights into the potential regulatory function of the ERVK solo-LTRs accumulated in Passerida, we collected the chromatin immunoprecipitation sequencing (ChIP-Seq) data of brain tissue of the zebra finch to annotate potential regulatory elements, including H3K27ac-, H3K4me3-, and H3K27me3-marked histone signals (Supplementary Data 4). Although we did not observe any significant signals of enrichment of ERVK solo-LTRs on regulatory regions under the

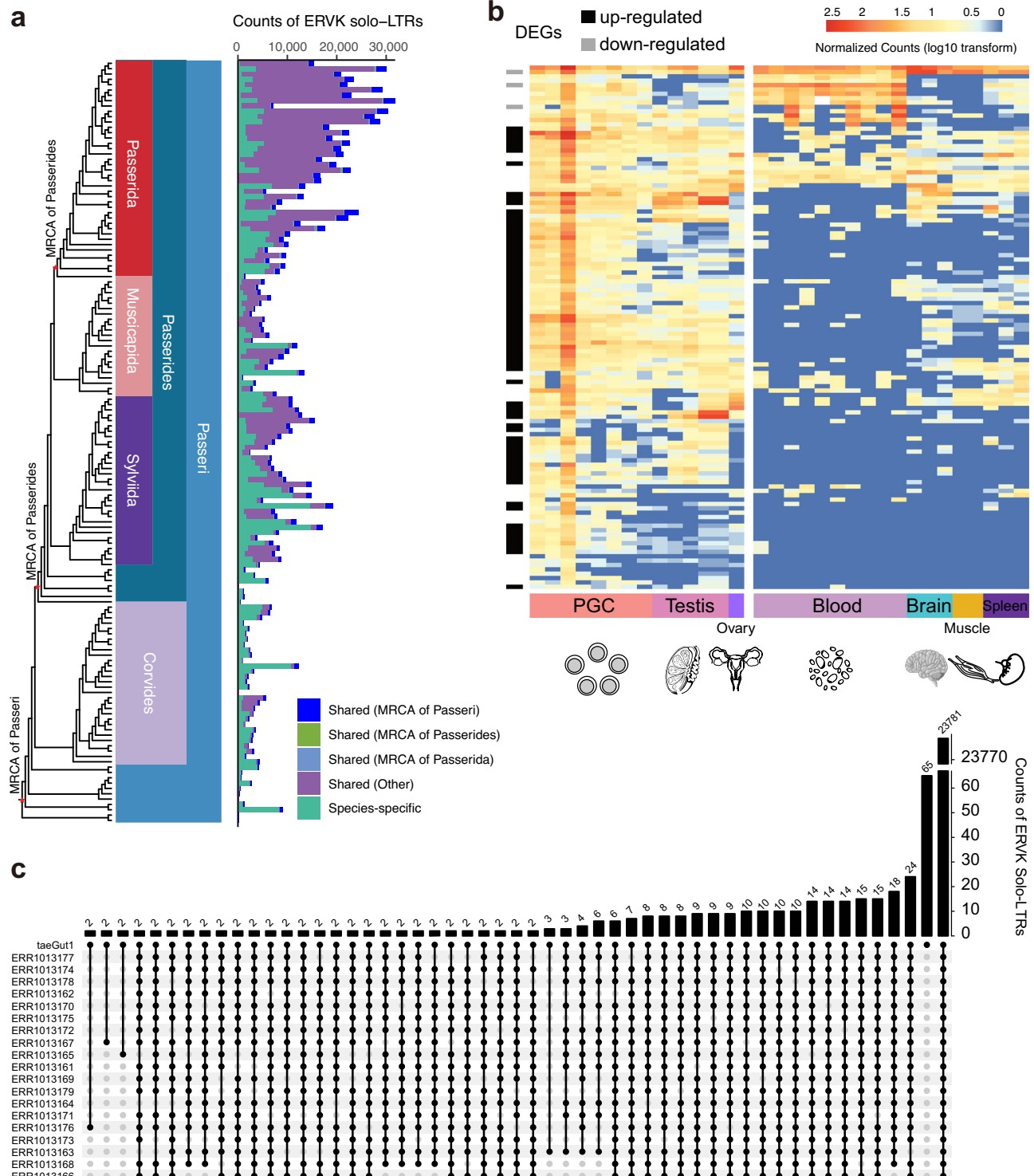

Fig. 3 | Indication of the ongoing accumulation of ERVK solo-LTRs. a Shared ERVK solo-LTRs constitute a large portion of all ERVK solo-LTRs in 41 Passerida bird species among 143 Passeri bird species. b Heatmap of the RNA expression levels of ERVK retroviral genes (n = 120) among three reproductive tissues and four other tissues. We applied a log10(count + 1) transformation to the normalized count for visualization. The black and gray cubes indicate significantly different levels of RNA expression between the ovary, testis, or primordial germ cells (PGC) in comparison

with other organs (thresholds: log₂ (fold change) > = 1 with Benjamini-Hochberg adjusted p-value < 0.05). c Upset plot illustrating the polymorphic status of ERVK solo-LTRs shared among the zebra finch population (n = 19). The top bars indicate the number of shared ERVK solo-LTRs, which are two or higher. Silhouettes of the testis, ovary, blood and brain are modified from the images by brgfx on www. freepik.com. Source data are provided as a Source Data file.

genome-wide context, we found a significant enrichment signal in ERVK solo-LTRs with ChIP-seq peaks compared to other types of ERV solo-LTRs (Fisher's one-sided exact test, p-value = 0.0028; Supplementary Data 5). Further, a total of 5535 ERVK solo-LTRs overlapped with regulatory regions and were located in the 10 kb flanking region of 1640 genes (Fig. 4a). These adjacent genes were significantly enriched in 43 GO terms, including the passive transmembrane transporter activity-related terms (GO:0022803, GO:0046873, GO:0022890, GO:0008324, and GO:0098662), channel activity-related terms (GO:0015267 and GO:0022836), blood circulation-related terms (GO:0008015 and GO:0003013) and the cellular components of neuron-related terms (GO:0150034 and GO:0043198) (Benjamini-Hochberg adjusted p-value < 0.05; Fig. 4b and Supplementary Data 6).

To determine whether newly acquired ERVK solo-LTRs in the zebra finch had become novel regulatory elements and were influencing the regulation of gene expression, we compared the expression patterns of 68 genes harboring new regulatory elements derived from ERVK solo-LTRs in the zebra finch to their orthologs in the chicken, which lacked these regulatory elements region. Using RNA-seq data from the brain tissue of the chicken and zebra finch, we found that 20 of the 68 genes were significantly differentially expressed between the two species. Specifically, in the zebra finch, 9 of these genes were up-regulated (*MC3R*, *NCOA3*, *UNC13B*, *ITGA2*, *ABHD10*, *CEP43*, *NMBR*, *MRM1*, and *ZBTB17*), and 11 were down-regulated (*SLC6A2*, *SLC39A11*, *NSDHL*, *BUB1B*, *CEP70*, *MREG*, *FAIM*, *EHHADH*, *RGS3*, *UBXN11*, and *BHMT*) (Fig. 4c).

Among the 11 down-regulated genes, the gene *SLC6A2* is used as the marker gene for the ventral tegmental area and locus coeruleus of the brainstem tegmentum in the zebra finch[29] (Supplementary Fig. 8), which encodes the norepinephrine transporter (NET) and regulates the reuptake of dopamine and norepinephrine into the presynaptic terminal of the synapses[30]. In a previous study, the decreased expression of *SLC6A2* in humans was reported to elevate norepinephrine levels and lead to a high heart rate in postural tachycardia syndrome (POTS)[31], which might be associated with the high resting heart rate (600–700 bpm) of the zebra finch[32].

Among the 9 up-regulated genes in the zebra finch, the *ITGA2* gene encodes the alpha subunit of a transmembrane receptor for collagens and related proteins. The expression of *ITGA2* was reported to be associated with the song rate (i.e., the number of songs during 10 min) in different morphs of the songbird, the white-throated sparrow (*Zonotrichia albicollis*)[33]. Likewise, the up-regulated expression of *ITGA2* in the brain of the zebra finch may be selectively advanced as individuals of this species rely on songs for social communication. We also found a 405-bp ERVK solo-LTRs insertion located in the upstream region of *ITGA2*, overlapping with the histone H3 on lysine 27 acetylation (H3K27ac) signal in the zebra finch, and potentially functioned as a *cis*-regulatory element to up-regulate the expression of the host gene (Fig. 4d). We conducted fluorescence in situ hybridization (FISH) for the *ITGA2* gene in the brains of the zebra finch and the chicken and confirmed a relatively higher expression of *ITGA2* in the brain of the zebra finch (Welch's two-sided t-test, p-value < 0.0001; Fig. 4e, f and Supplementary Fig. 9a, b). We further found that the expression of the *ITGA2* gene was varied in different brain regions of the zebra finch, with higher expression levels in the well-known brain regions containing the vocal motor pathway[34,35] – namely the HVC (used as a proper name)[36] and robust nucleus of the arcopallium (RA) regions – than the Area X regions (Fig. 4e).

To determine the potential *cis*-regulation function of the 405-bp ERVK solo-LTR, we first searched the motifs using the program findM[37], and found 6 Initiator (Inr) core promoter significant signals in the 405 bp insertion fragment. Given that the initiator was one of the four core promoter motifs[38], it was very likely that this 405-bp ERVK solo-LTR had the ability to initiate the transcription of downstream genes.

To verify this, a dual luciferase reporter assay experiment was designed in the chicken fibroblast cell line (UMNSAH/DF-1). The 405-bp ERVK solo-LTRs were ligated into the control vector (pGL3-Basic) (Promega) upstream of the firefly luciferase (*Luc*) gene as the pGL3-Promoter vector (Supplementary Fig. 9c). The internal control pRL-TK Renilla vector (Promega) was co-transfected with the pGL3-Promoter vector or pGL3-Basic empty vector, as the experimental and control group, respectively. After normalizing with the internal control Renilla luciferase activity, we detected a significantly higher relative luciferase activity in the experimental group than in the control group (Welch's two-sided t-test, p-value < 0.0001; Fig. 4g), confirming that the 405-bp residue of the ERVK solo-LTRs was indeed capable of introducing *cis*-regulatory activity to up-regulate the expression of a downstream gene. Given the relatively high expression of the *ITGA2* gene in the HVC and RA regions, we speculate that the retention and preservation of ERVK solo-LTRs may have contributed to the evolution of the vocal learning-related brain region through the provision of transmembrane activity.

## Discussion

Endogenous retroviruses (ERVs) have been co-opted by their hosts throughout long-term co-evolutionary processes to serve a variety of biological functions, such as cell differentiation (syncytins in trophoblast fusion and differentiation)[16], the regulation of the immune system[25], and early embryo development[39]. The activity of ERVs also impacts the health of their hosts[15]. The LTRs of ERVs often carry the primary promoter and regulatory elements for provirus expression[22], and may also function as *cis*-regulators to their hosts in the form of solo-LTRs. Despite the widespread distribution of solo-LTRs in the genomes of different species, the biological function of solo-LTR formation remains poorly known. Different from the previous hypothesis that a mechanism resisting viral invasion into the genome might have been evolved in bird species due to the adaptation to flight lifestyle[10,11], our study shows that at least in Passeriformes, which comprise over 60% of modern bird species, ERVK is highly active and has continuously expanded throughout the diversification of this group. We propose that in Passerida species, the formation of solo-LTRs may serve as a host defense mechanism for purging newly inserted ERVK or as a consequence of the high recombination activity among the LTRs, thereby counteracting the deleterious consequences of the expansion of ERVK in this group. Additionally, we show that several solo-LTRs in Passerida species carry regulatory elements that introduce potential novel regulatory interactions and may contribute to trait variation, thereby facilitating natural selection.

By studying the genomes of over 300 bird species, our study unveiled the burst of ERVK throughout the diversification of Passeriformes, especially in the parvorder Passerida after Paleogene–Neogene boundary (around 22.4 MYA). Previous studies have identified ERVK-like sequences across a diverse range of organisms, including primates, rodents, ungulates, fish, and insects[40,41]. It is interesting to note that the invasion of ERVK in the genomes of some of these taxa occurred during the more recent radiation[42–44], which occurred in geological periods prior to the Passeriformes radiation. The integration of ERVK in human genomes could be traced back to the ancestor of Catarrhine primates[45], and the bursts of ERVK amplification occurred in the ancestor of Hominoidea (near the Paleogene–Neogene boundary), Homininae and human, respectively[44]. A high proportion of ERVK has also been reported in Muridae rodents which originated in the early Neogene[42,43]. The convergent bursts of ERVK in the genomes of several taxa imply that paleoviruses of ERVK might have spread throughout continents where these animals were distributed, and integrated into their hosts' genomes frequently during the Paleogene–Neogene boundary. However, ERVK has experienced different evolutionary fates in the genomes of

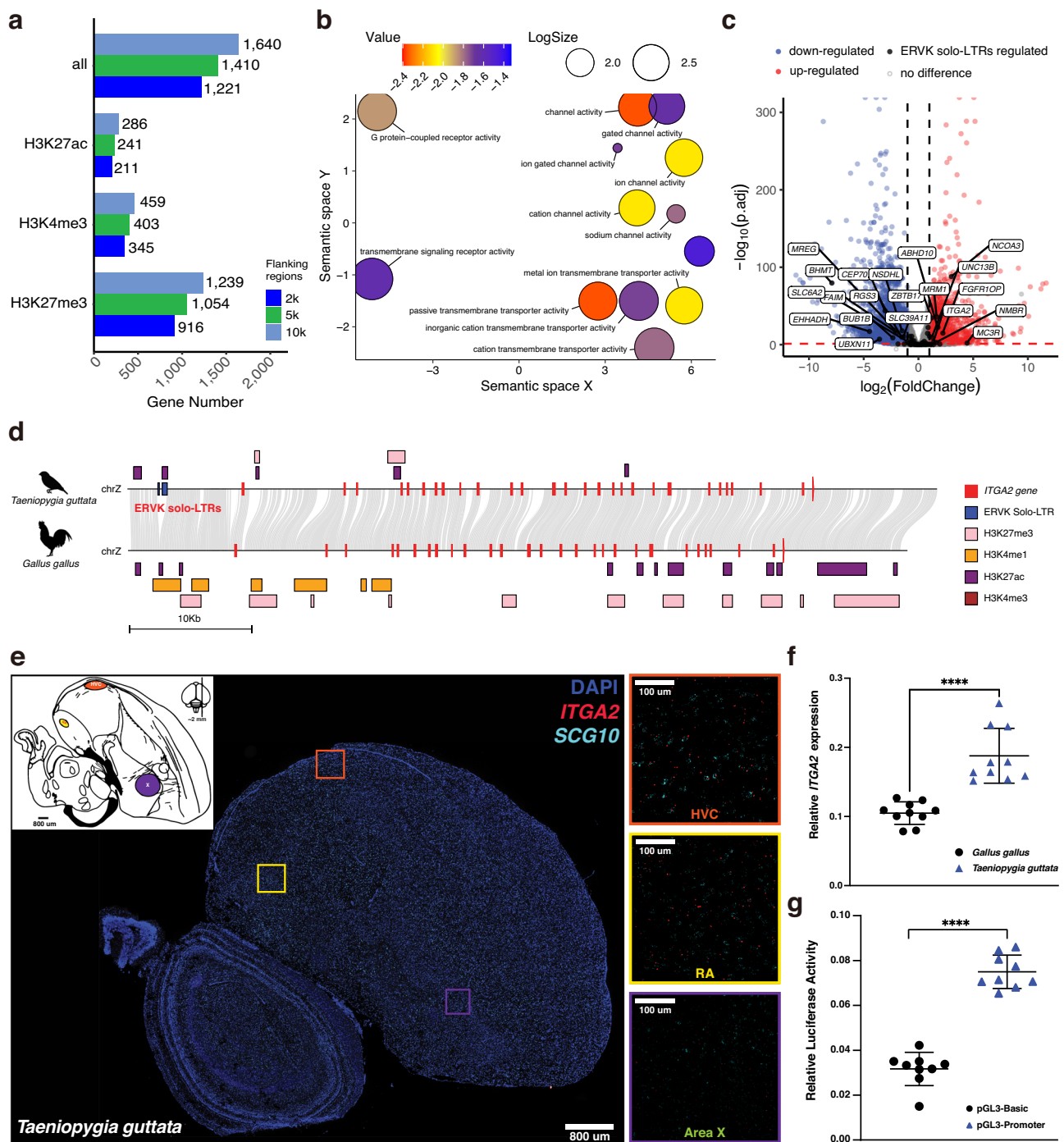

**Fig. 4 | ERVK solo-LTRs function as regulatory elements in the brain of the zebra finch. a** In zebra finch, the number of genes located within 2 kb, 5 kb, and 10 kb flanking regions of the regulatory elements overlap with 5535 ERVK solo-LTRs. **b** Significantly enriched GO terms of 1640 genes on molecular function were visualized by REVIGO tool. **c** Volcano plot shows DEGs between the brain tissues of zebra finch and chicken among 12,028 orthologous genes. Horizontal red dashed line indicates the cutoff of Benjamini-Hochberg adjusted *p*-value = 0.05, and the vertical black dashed lines indicate a 2-fold change. **d** Genomic collinearity plot of *ITGA2* gene showed that the solo-LTRs residue overlapped with H3K27ac signal in zebra finch. Silhouettes of the zebra finch and chicken are from https://www.phylopic.org/. **e** Fluorescence in situ hybridization (FISH) microscope photographs of the brain of a zebra finch. Expression signals of *ITGA2* and *SCG10* were indicated in red and cyan, respectively. The *SCG10* gene, which encodes a neuron-specific stathmin protein, was used as control. The blue signal, obtained from 4' 6-diamidino-2-phenylindole (DAPI) staining, shows the location of the nucleus. The

HVC, RA, and Area X regions are labeled according to the schematic diagram, which was modified based on the data retrieved from the ZEBrA database (Oregon Health & Science University, Portland, OR 97239; http://www.zebrafinchatlas.org)[29]. **f** Relative *ITGA2* expression to the DAPI signal revealed that *ITGA2* was highly expressed in the zebra finch compared to the chicken. The *ITGA2* and DAPI signals were measured using the "Analytical Particle" function of the software FIJI[82], with ten random samples (*n* = 10) taken from regions throughout the whole brain. **g** Dual-luciferase reporter assay in the UMNSAH/DF-1 cell line demonstrated the potential *cis*-regulation activity of the 405 bp ERVK solo-LTRs insertion located upstream of the *ITGA2* gene in the zebra finch. Relative luciferase activity was determined with *n* = 9 biologically independent repetitions for each experimental group. Horizontal lines indicate the mean (± s.d.) in (**f, g**). Welch's two-tailed t-test was conducted using GraphPad Prism (*p*-value: **** < 0.0001) in (**f, g**). Source data are provided as a Source Data file.

mammals and birds. Throughout the evolution of mammalian species, ERVK has constantly accumulated in the genome. Some ERVs, such as HERV-K, still retain the capacity to encode viral proteins and form virus-like particles that participate in several cellular processes during the embryogenesis of the host[46]. The constant activity of HERV-K can also elicit senescence in young cells, resulting in more extracellular retrovirus-like particles and activation of the innate immune system[15]. In contrast, throughout the evolution of birds, both the potential selection pressures to purge ERVs and high recombination activity among the LTR might have resulted in the intervening proviral sequences for most avian ERVKs being eliminated through a substantially higher frequency of ERVK solo-LTR formation. Moreover, the expansion of ERVK solo-LTRs in Passerida bird species is still an ongoing process, marked by high transcriptional activity of ERVK retroviral genes in the reproductive organs and a high diversity of ERVK solo-LTRs among populations of zebra finches. Thus, the formation of solo-LTRs may continually act as a mechanism in birds to suppress the deleterious effects of the concurrent ERVK expansion. The activity of ERVK has also been reported to increase human population diversity and inferred to contribute to the divergence between humans and chimpanzees[47]. Recent studies have highlighted associations between ERVK activity and various human diseases[19,48]. For instance, the activity of ERVK may increase somatic mutation rate[49], especially in cancer cells[50]. Research has also established the significance of ERVK to aging; in both mice and humans, the aberrant activation of ERVKs induced cellular senescence and accelerated tissue aging[15]. The formation of solo-LTRs to remove the main functional bodies of ERVs may therefore function as an efficient strategy to alleviate the mutational loads introduced by the ongoing activity of ERVs to host species[51]. As more high-quality avian genomes become available[52], we will be able to further investigate the evolutionary effects of efficient EVEs purging on host health and the genomic landscape.

Studies on plants and animals have documented the co-option of regulatory functions of ERVs LTRs[53,54]. Other studies have shown that solo-LTRs may act as promoters of adjacent genes either in the sense or the antisense[24]. For instance, the *TAp63* gene in Hominidae co-opted the LTRs of ERV-9 as a promoter upstream from the gene, which could initiate testis-specific *TAp63* transcription. This co-option was associated with the higher fitness and longer reproductive periods in Hominidae[55]. Contributing to this literature, our study provides experimental evidence to suggest that ERVK solo-LTRs may function as *cis*-regulatory elements, as demonstrated in the case of the *ITGA2* gene. In the mammalian genome, a high frequency of ERVK was located near the human-specific topological associating domains (TADs)[56]. In a similar vein, our results show that ERVK solo-LTRs in the zebra finch likewise display significant enrichment with ChIP-seq signals compared to other ERVs. The expansion of Passeriformes-specific ERVK solo-LTRs can supply novel genetic materials as regulatory elements that shape adaptive evolution[1]. Interestingly, the recent expansion of ERVK in Passeriformes was enriched in genes related to transmembrane activity, which could lead to more efficient signal transmission in the birds' brains and further enhance their vocal-learning abilities. Passeriformes species are known for their exceptional diversification and vocal learning capabilities. The retention of ERVK solo-LTRs in the genomes of Passeriformes species likely reflects the outcome of natural selection, which acts to preserve the new regulatory regions associated with advanced traits and drives the adaptive evolution of species into specialized ecological niches[57]. Overall, our findings suggest that the emergence of novel regulatory elements resulting from the burst of ERVK solo-LTRs in the genomes of Passeriformes species may provide abundant lineage-specific regulatory machinery, which has in turn facilitated the impressive diversification of these birds, and provide empirical support for the gene regulation aspect of the TE-thrust speciation hypothesis[58].

## Methods

### Ethics declarations

All presented experiments and procedures were approved by the Animal Use and Care Committee of Zhejiang University following the Guidelines of the Care and Use of Laboratory Animals in China.

### Genome and phylogenetic data

Using the Bird 10,000 genome project (B10K) and the NCBI database, we collected data on the genomes of 405 vertebrate taxa representative of the amniotes; this included data for 362 bird species, 23 reptilian species, and 20 mammalian species (Supplementary Data 1). We derived a phylogenetic tree for the 362 bird species from the B10K family-level ASTRAL tree[59] and the phylogenetic tree for 10,135 bird species (from the supplementary data of Feng et al.[7] and Brown et al.[60]). We collected data on the phylogenetic relationships between reptiles and mammals from the website: http://www.timetree.org and made the following four modifications: *Myanophis thanlyinensis* classified under Homalopsidae; *Python bivittatus* replaced with *Python molurus*; *Gopherus evgoodei* replaced with *Gopherus agassizii*; *Canis familiaris* replaced with *Canis lupus*. We then pruned the lengths of branches on each synthesized tree for birds, reptiles, and mammals, and visualized all trees using the R package *ggtree* (v3.3.0.900)[61].

### Identifying solo-LTRs

We identified solo-LTRs from the genomes of 405 representative amniote species (including 362 birds, 23 reptiles, and 20 mammals) following the pipeline presented in Supplementary Fig. 1.

a). Build the LTRs database: we used RepeatMasker (v4.1.2)[62] with the RepBase library (v.20170127) and LTR-harvest (v1.6.1)[63] implemented in GenomeTools to identify candidate regions for long terminal repeat (LTR) sequences for each genome. Then, we obtained the intersection of these two datasets of LTR sequence regions for each genome, by applying the function "intersect" of the bedtools package (v2.30.0)[64] within [100, 1000] bp[63]. We combined the intersection LTR sequences of birds, reptiles, and mammals respectively, and applied the software cd-hit (v4.8.1)[65] with with a cut-off of 95% similarity to reduce redundancy and to form the non-redundancy LTR datasets for three groups.

b). Identify the LTRs region: we employed blastn (v2.9.0)[66] using the parameters "-prec_indetity 65 -qcov_hsp_perc 80" with the corresponding non-redundancy LTR sequences database as the query to re-blast each species' genome. Next, using the bedtools to cluster redundant regions of LTRs based on coordinates, we retained the best identified region (according to the scores and e-value from blastn) as the non-redundant LTRs region.

c). Identify the paired-LTRs and solo-LTRs: To we identified the pairs of LTRs within distances of 20 kb using the reciprocal-best-hit (RBH) method with the threshold of 85% identical (other threshold were shown in Supplementary Note, Supplementary Fig. 10 and Supplementary Data 7), and identified the remaining unpaired LTRs as solo-LTRs. To avoid the deviation of assembly quality, we have filtered all LTRs located on scaffolds with a length of less than 20 kb. The script for solo-LTR identification in this study is available on GitHub (https://github.com/ChenGuangji/BirdsSoloLTRs).

Considering the removal process of transposons would lead to the target site duplications (TSDs) being imperfect and generating sequence diversity[67], it might cause the specific boundaries of TSDs and LTRs to not be so clear[68]. To address this, we not only checked the target site duplications (TSDs) in the proximal 10 bp upstream and downstream for the paired-LTRs (intact LTRs) and solo-LTRs as employed in Peona et al.[69], but also extended the flanking region into 15 bp and 20 bp. Using blast with the criterion 4 to 6 bp complete hits by following the method from Peona et al.[69], the patterns of solo-LTRs formation between birds and other species were still consistent across different criteria (details were recorded in the Supplementary Note,

Supplementary Figs. 11–15, Supplementary Data 8, and Supplementary Data 9).

### Tracing insertions of ERVK solo-LTRs

We obtained the reference-free whole genome cactus alignment of all bird species from Feng et al.[7] and Armstrong et al.[70]. Using a parallelized version of the command hal2maf (https://github.com/ComparativeGenomicsToolkit/hal), we extracted each species' ERVK solo-LTRs region from the reference-free whole genome cactus alignment against other Passeri (Oscines) species. For each species, we defined any ERVK solo-LTRs region which was present in an alignment block in another Passeri species as "Shared", and the remaining ERVK solo-LTRs region as "Species-specific." Further, any "Shared" ERVK solo-LTRs region that aligned with three ancestor notes (MRCA of Passeri, MRCA of passerides, and MRCA of Passerida) was subdivided, respectively.

### Investigating the accumulation of ERVK solo-LTRs in the zebra finch

**Using RNA-Seq data to reveal the expression of ERVK retroviral genes in sexual organs.** We collected RNA-seq data from seven organs (including primordial germ cells, testis, ovary, blood, brain, muscle, and spleen) of the zebra finch from a public database (Supplementary Data 4). Ribosomal RNA reads were removed from these quality-trimmed RNA-Seq reads by employing KneadData (v0.10.0) according to the SILVA ribosomal RNA database (v0.2). The cleaned RNAseq reads were then aligned to the zebra finch's reference genome using Hisat2 (v.2.2.1)[71], followed by quantification using "htseq-count" from the HTSeq package (v.0.13.5)[72] with the annotations for protein-coding genes obtained from Feng et al.[7]. Output from htseq-count was imported to DESeq2 (v.1.34.0)[73] for normalization and differential expression analysis. In the differential expression analysis, an absolute value of $\log_2$ (fold change) $>= 1$ and an Benjamini-Hochberg adjusted $p$-value $< 0.05$ were set as thresholds for differential expression.

**Uncovering polymorphisms of ERVK solo-LTRs at the population level with whole genome re-sequencing data.** We collected the whole genome re-sequencing data for 19 individual zebra finches from a public database (Supplementary Data 4). Using BWA-MEM[74] with default settings, reads were mapped to the reference genome of the zebra finch (GCA_000151805.2). We used the coverages of the ERVK solo-LTRs identified on the reference genome in each individual as an indicator to calculate the shared states of each ERVK solo-LTR. The final shared state was visualized the intersections (intersection size $>= 2$) with the R package UpSetR (v.1.4.0)[75].

### GO enrichment analysis

First, we used the eggNOG-mapper[76] to annotate the Gene Ontology (GO) annotation information of protein-coding genes of the zebra finch obtained from Feng et al.[7] with the default parameters. Second, we built the GO annotation as the local R package of zebra finches with the R package AnnotationForge (version 1.42.0)[77]. Then, we applied the GO enrichment analysis by applying the "enrichGO" function of the R package clusterProfiler (version 4.6.2)[78] with the Benjamini-Hochberg adjusted $p$-value $< 0.05$ (Supplementary Data 6), for genes harboring the ChIP-Seq peak overlapping with ERVK solo-LTRs in 2 kb, 5 kb, 10 kb flanking region. Finally, we used the REVIGO tool (available at http://revigo.irb.hr/) to eliminate redundancy and perform the visualization for the result base on the 10 kb flanking region.

### Annotation of the promoter motif

We searched for the promoter motif of the 405 bp ERVK solo-LTRs located in the upstream of the *ITGA2* gene in the zebra finch using the FindM tool (http://ccg.vital-it.ch/ssa/findm.php) and the promoter motif library, applying a $p$-value $< 0.01$ as a cutoff for statistical significance[37].

### Fluorescence in situ hybridization

We used four adult zebra finches (*Taeniopygia guttata*, age 2–3 months) and four adult chickens (*Gallus gallus*, age 2–3 months) for fluorescence in situ hybridization (FISH). All animal handling procedures were approved by the Animal Use and Care Committee of Zhejiang University following the Guidelines of the Care and Use of Laboratory Animals in China. All efforts were made to minimize the number of animals used and their suffering. For FISH, fresh brains were quickly removed and washed gently in DPBS (DEPC-treated PBS). The brains were cryosectioned in the sagittal plane at 10 μm using a freezing microtome machine (Leica CM1950, Germany), and sections were mounted to charged slides and stored at −80 °C until use. DNA probes of *ITGA2* and *SCG10* were designed using an amplification-based single molecule FISH (asmFISH) DNA ligation probe (DLP) designing principle[79]. In accordance with previously described experimental and detection methods[80,81], asmFISH was performed with the brain sections of the zebra finch (between 2.00 and 2.25 mm lateral to the midline) and chicken (the corresponding region between 3.40 and 3.90 mm lateral to the midline). The images were acquired with a Leica DM6B fluorescence microscope (Leica, Germany) using a 20 × objective and exported in TIFF format. Further image processing was conducted using FIJI[82] (https://imagej.net/Fiji), where regions of interest (ROIs) were manually drawn with ten random samples across the whole brain region, with the aid of the Zebra Finch Expression Brain Atlas (http://www.zebrafinchatlas.org/) as a reference[29].

### Dual-luciferase reporter assay

Chicken fibroblast cell lines (UMNSAH/DF-1) were maintained at 38 °C in an incubator with a humidified atmosphere of 5% $CO_2$. The fragment of ERVK solo-LTRs with potential *cis*-regulatory element activity (405 bp) was chemically synthesized and digested with *Xho*I and *Nhe*I (New England Biolabs, Ipswich, MA). The target DNA fragment was cloned to the pGL3-Basic vector (Promega) in the presence of a DNA Ligation Kit (Takara, Japan) as a pGL3-Promoter. Cells were prepared in 12-well plates and the pGL3-Promoter and pRL-TK vector (Promega) were co-transfected to cells using Lipofectamine 3000 reagent (Invitrogen, USA) (in accordance with the manufacturer's instructions), with pGL3-Basic empty vector as a control. Forty-eight hours after transfection, a luciferase reporter assay was conducted to measure potential *cis*-regulation activity using the Dual-Luciferase Reporter Assay System (E1910, Promega, USA). Firefly and Renilla luciferase activities were measured using a GloMax 96 Microplate Luminometer (Promega, USA). The Firefly/Renilla signal ratios were recorded as relative luciferase activity. Using GraphPad Prism software, a statistical analysis was then performed with the Welch's two-sided t-test between the pGL3-Basic and pGL3-Promoter groups.

### Reporting summary

Further information on research design is available in the Nature Portfolio Reporting Summary linked to this article.

## Data availability

The 362 avian genome assemblies and annotations data are accessible through the CNSA public database (https://db.cngb.org/cnsa/) with accession number CNP0000505 and NCBI database with accession PRJNA545868. The other 42 genome assemblies, DNA sequencing and RNA-seq data used in this study can be found in the NCBI public database according to Supplementary Data 1 and Supplementary Data 4. Source data are provided as Source Data files and are available on Zenodo (https://doi.org/10.5281/zenodo.10812365). Source data are provided with this paper.

## Code availability

All code underlying these analyses and statistics in this study is available on GitHub (https://github.com/ChenGuangji/BirdsSoloLTRs) and Zenodo (https://doi.org/10.5281/zenodo.10812365).

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

## Acknowledgements

We thank Hao Huang (Hangzhou Normal University), Guo Ding (Zhejiang University), Zheyi Ni (Zhejiang University), Yang Zhou (BGI), Pei Zhang (BGI), and Qi Fang (BGI) for helpful discussions on the experiment and analysis, Qiong Huang, Dan Yang, and Jingyao Chen (Core Facilities, Zhejiang University School of Medicine) for their technical support, and China National GeneBank for providing the computation resource. This work was supported by a National Natural Science Foundation of China grant (no. 32170626) and a National Key Research and Development Program of China grant (no. 2023YFA1800500) to S.F. This work was also supported by the Strategic Priority Research Program of the Chinese Academy of Sciences (XDB31020000), the International

Partnership Program of Chinese Academy of Sciences (no. 152453KYSB20170002) to G.Z. This work was also supported by Shanghai Municipal Science and Technology Major Project (ZD2021CY001) and a grant from the CAS Interdisciplinary Innovation Team to J.C. This work was also supported by the Natural Science Foundation of Fujian Province (2022J06022) to R.K.

## Author contributions

S.F. designed the study. S.F. and G.Z. supervised all analyses. G.C., X.L., X.W., S.F. and J.C. contributed the solo-LTRs identification. G.C. performed the solo-LTRs identification, the correlation statistical analyses, and analysis of transcriptome and resequencing data. G.C., D.Y., Y.Y., D.S., Y.L. and R.K. contributed and performed the experiments. G.C. and D.Y. performed the statistical analysis of experiments. G.C., S.F. and G.Z. wrote the manuscript with input from all authors. All authors approved the manuscript before submission.

## Competing interests

The authors declare no competing interests.

## Additional information

[1]College of Life Sciences, University of Chinese Academy of Sciences, Beijing, China. [2]Center for Evolutionary & Organismal Biology, Liangzhu Laboratory, Zhejiang University School of Medicine, Hangzhou, China. [3]BGI Research, Wuhan, China. [4]Center for Genomic Research, International Institutes of Medicine, The Fourth Affiliated Hospital, Zhejiang University School of Medicine, Yiwu, Zhejiang, China. [5]School of Medicine, Huaqiao University, Xiamen, Fujian 361021, China. [6]CAS Key Laboratory of Molecular Virology & Immunology, Shanghai Institute of Immunity and Infection, Chinese Academy of Sciences, Shanghai, China. [7]Innovation Center of Yangtze River Delta, Zhejiang University, Jiashan, China. [8]Department of Infectious Diseases, National Medical Center for Infectious Diseases, Huashan Hospital, Institute of Infection and Health Research, Fudan University, Shanghai, China. [9]Laboratory for Marine Biology and Biotechnology, Qingdao Marine Science and Technology Center, Qingdao, China. [10]Shanghai Sci-Tech Inno Center for Infection & Immunity, Shanghai 200052, China. [11]Shanghai Key Laboratory of Infectious Diseases and Biosafety Emergency Response, Huashan Hospital, Fudan University, Shanghai, China. [12]Department of General Surgery of Sir Run Run Shaw Hospital, Zhejiang University School of Medicine, Hangzhou, China. ✉e-mail: jiecui@fudan.edu.cn; fengshaohong@zju.edu.cn

