## [Peer Review File · Nature Communications]

Adaptive expansion of ERVK solo-LTRs is associated with
Passeriformes speciation eventsREVIEWER COMMENTS

Reviewer #1 (Remarks to the Author):

Guangji Chen et al. "Adaptive expansion of ERVK solo-LTRs in 1 accompanying with Passeriformes speciation"

In this study, the authors investigate the evolution of endogenous retroviruses (ERVs) in birds, and more specifically the solo-LTRs (long terminal repeats) formed by the purging endogenous viral elements (EVEs) from the genome. LTRs make up a very large proportion of the genomes in many animals and plants, but are rarely studied in birds in general. Passerine birds constitute an extraordinary exception from this, and the authors explore the extent and explanation of this based on 362 bird genomes generated from the B10K project.

The study is well-designed and very well conducted. The results are highly interesting and discussed in relation to our previous understanding of these processes. I am not an expert on the evolution of ERVs but it seems to me that the referred literature is both appropriate and exhaustive. My comments are restricted to questioning certain claims by the authors, as well as a few typos and minor details.

Line 126: The authors observe a high frequency of solo-LTRs formation in birds, indicating a strong selection against EVEs. They then suggest that this show that birds may have experienced lower rates of paleoviral infiltrations. It is unclear to me why this must be the case. Birds clearly seem efficient in purging EVEs from the genome, but how does this relate to the rate of paleoviral infiltrations?

Line 131: Change "neo-ave" to either "neoavian" or "Neoaves".

Line 132: "TE" – is this abbreviation explained somewhere in the text?

Line 133 ff.: The authors found a "notably higher" proportion of solo-LTRs in Piciformes and Bucerotiformes but "these were not significantly higher" in other species. To me, the observed non-significance suggests that one cannot say they are higher in these two groups.

Line 153: avoid using "significant" if not describing the results of a statistical test.

Line 155 ff.: The comparisons in the text and in Fig. 2b of the authors present statistics for Passeriformes, Passeri, Passerides and Passerida. The figure 2b shows that there is very little difference between these statistics. This is expected since all lower taxa constitute a subsample of the next higher taxon. As the general point is to show the marked expansion of ERVK solo-LTRs in passerines I think it is enough to discuss and show the statistics for Passeriformes.

Line 155: The supplementary figure 3 shows that not only ERVK shows the expansion in passerines, this seems true also for ERV1.

Line 161: delete "birds", change "has" to "was".

Line 164, supplementary figure 4: The figure shows a positive correlation also for Corvides for ERVL and ERV1.

Line 164, supplementary figure 4: The x-axis shows number of splits along the phylogenetic lineages. My question is if the number of splits is used as proxy for time, so that the x-axis in reality shows time. Or, if it actually is the number of splits that correlates with the proportion of LTRs, i.e. that the recombination that occurs at these splits somehow is correlated with the proportion of LTRs? Please address this some place in the text or in the figure legend.

Line 166: shouldn't this be "EVRK solo-LTRs"?

Line 167. The p-value for the correlation seems extremely low. The same is true for those in fig. 2b and even more so in suppl. fig. 5. Please check. Also, Pearson's correlation coefficient is "r" (not capital R).

Line 203-204: It could well be due to my lack of solid knowledge about this matter, but the sentence that starts with "Such higher..." is unclear to me. Please check if further clarification is needed.

Line 213: Fig. 3c: It says the top bars indicate shared LTRs that are higher than two, shouldn't it be "two or higher"

Line 321: It seems the sentence that starts with "Our study..." should be part of the previous sentence and separated by a comma.

Line 334: I suggest replacing "proximal" with "prior".

Line 321 "Our study" should be "our study"

Line 386: Change "1000" to 10,000".

Line 398: It is unclear to me if the identification of candidate region that is described in this section is done only for the zebra finch or not. Please clarify.

Supplementary Fig. 7: It seems as the graph is truncated at -2 at the y-axis? If it is I believe it would be better to show the whole figure.

Reviewer #2 (Remarks to the Author):

The authors present an extensive analysis of the evolution of LTR retrotransposons throughout the avian phylogeny using the large genomic dataset provided by the B10K project. The authors investigate the presence of species-specific LTRs, their polymorphisms and their diversity in the different species. Thanks to the use of transcriptomic data on top, the authors are able to measure the transcription of LTRs and genes in several tissues including primordial germ cells. They were able to identify some candidate LTRs that may serve as gene promoters and test the functionality of one of them in vitro finding positive evidence for its function as promoter. I find the analysis very interesting and relevant to the field, but I have some minor and major points to address.

Minor

- 1) Line 49-52: I would rephrase more generally in "which results in the deletion of the internal region" so to include all those instances of truncated, non -autonomous ERVs that lost any viral ORF.
- 2) Line 72: I would rephrase in "the widely studied syncytin1 gene, derived from the HERV-W env".
- 3) Line 93-96: This is a strong statement since more neutral processes can lead to the same pattern, like the higher frequency of recombination in avian genomes compared to mammals. On top of that, the rates of homologous recombination and non-allelic homologous recombination - responsible for the formation of solo LTRs - are associated as they are part of the same molecular phenomenon. Selection has not been tested in any on the analysis therefore I would highly recommend the authors to stick to more neutral explanations and add selection only as a speculation.
- 4) Line 188-189: I find the phrasing here very confusing. Passerida still have around 10% of genome repetitive, how is it possible that solo LTRs make up 76% of the genome?
- 5) Line 192 and throughout the manuscript: the abbreviations should be introduced in parentheses after their complete form outside the parentheses.
- 6) Line 203-205: the reproductive organs are very heterogeneous tissues therefore expression in the whole organs is not necessarily a prerequisite for integration in the next generation. Activity in the primordial germ cells can be. Please rephrase.
- 7) Figure 3: I would not say that the transcription of ERVs is evidence for their ongoing accumulation, it is a necessity but not sufficient for their transposition and as evidence for their ongoing integration, therefore please change the caption of the figure.

Major

- 1) Line 230-231: 10 kb is a rather large threshold, what does it happen with smaller thresholds? These thresholds are completely arbitrary so it would be good to explore the effects of different thresholds on your results.
- 2) I find the description of the method to identify solo LTRs is not clear enough in its steps even in the supplementary, so I kindly ask the authors to make it clearer as it is the foundation for the entire manuscript.
- 3) As far as I understand, I do not really see in the method a step in which the authors distinguish solo LTRs from fragmented LTR retroelements. I think full-length elements are masked away from potential solo LTRs in the first step of the pipeline with LTRharvest but I do not think this is sufficient for a couple of reasons: 1) the genomes used from the B10K project are highly fragmented which hinders drastically the recovery of full-length elements because not assembled at all thus biasing the counts; 2) fragmented LTR retroelements that can result from independent deletion events and/or internally truncated elements will not be clearly identified by LTRharvest since they lost the protein coding genes. Both points will inflate the number of solo LTRs in comparison to full-length and inflate the number of false positives. I suggest following methods in which target site duplications at the extremities of the potential solo LTR are taken into consideration like in <https://doi.org/10.1101/2021.12.31.473444>

Reviewer #3 (Remarks to the Author):

Review of Chen et al., "Adaptive expansion of ERVK solo-LTRs in accompanying with Passeriformes speciation" for Nature Communications

The paper by Chen et al. is an interesting application of large-scale comparative genomics to assess potential mechanisms shaping evolution. In this case, the mechanism of interest is the generation of ERV solo-LTRs, with their potential to regulate the expression of adjacent genes.

Here, they first show that Solo-LTR numbers are positively correlated with genome size in mammals and reptiles, but not in birds. They interpret this as evidence for "strong selective pressures to purge ERVs from bird genomes." This is not exactly a novel observation (since we already knew that birds have smaller genome sizes and reduced repetitive sequence content), but it does advance the field by highlighting a specific mechanism that may be especially active in birds for pushing back against transposon-mediate genome size expansion. (But why this is so in birds remains a mystery!).

Next, they show that ERVK solo-LTRs, in particular, appear to increase during more recent speciation events, especially within Passeriformes. Indeed, almost half of ERVK solo-LTRs were species-specific in the suborder Passeri and appear to have emerged through "multiple recent bursts in a lineage-specific way." Moreover, using whole genome re-sequencing data from 19 zebra finch individuals, they found that 655 (2.68%) ERVK solo-LTRs were polymorphic among individuals, suggesting an active reservoir of recent (and probably ongoing) genetic variation. This is not a radical discovery in general (retrotransposons are active!), but it's nice to see this direct and detailed characterization in songbird lineages and the zebra finch.

To assess the potential functional impact of ERVK insertions, they mined public RNAseq data for the zebra finch, finding evidence for transcriptional activity among 120 ERVK genes. They looked at 68 genes that harbored new ERVK solo-LTs in zebra finch relative to chicken and found that 20 were differentially expressed between the two species. They then focused on the ITAG2 gene, building on prior evidence that its expression is correlated with song rate in another songbird species (white-throated sparrow). Looking in the zebra finch, they find an ERVK solo-LTR in the ITAG2 upstream region overlapping with an H3K27ac ChIP-Seq signal. From this, they developed a plausible (but still "just so") story suggesting how this flanking ERVK solo-LTR might support the evolution of singing behavior. Indirectly supporting this, they transfected this element into chicken fibroblasts and found evidence that it can indeed increase the expression of a flanking reporter gene.

I don't see any major flaws in the analysis and presentation, though I would direct the authors to the following specific details and considerations.

Line 37 and elsewhere: HVC is referred to as an acronym for "High Vocal Center." To my knowledge, no one has defined HVC as an acronym since the nomenclature was revised in 2004 (Reiner et al., "Songbirds and the Revised Avian Brain Nomenclature". *Annals of the New York Academy of Sciences*. 1016 (1): 77–108. doi:10.1196/annals.1298.013). Rather, the community seems to favor "HVC (used as a proper name)" and then citing the Reiner paper.

Line 145: "Bird species with potentially problematic assemblies (genome size < 800 Mb or scaffold N50 < 20 kb) were filtered to reduce the bias of assembly quality." I assume they mean they removed these assemblies from the analysis? But this actually raises another consideration: Might there be some systematic difference in the way the Bird 10K assemblies were assembled with respect to repetitive sequence content compared to the non-avian assemblies? For example, how many assemblies were based on short-read vs long-read sequencing technologies? Is there a reason we can discount this as a possible confounder in the analysis?

Line 161 (missing word): "The speciation process of Passeriformes birds has BEEN accompanied by

an accumulation of ERVK solo-LTRs.”

Lines 240-245: if you just picked any 68 genes at random, what proportion would be differentially expressed between ZF and chicken? (Knowing this would confirm the statistical significance of their observation that about a third of ERVK-associated genes are differentially expressed).

Line 256: is there evidence for the flanking ERVK solo-LTR in the genome of the white-throated sparrow (*Zonotrichia albicollis*)?

Lines 320-321: an incomplete sentence.

Line 324: “The formation of solo-LTRs may serve as a host defense mechanism for purging newly inserted ERVK” -- this implies a discrete mechanism focused on defense. That would be interesting if true. But is there any support for this as a defense mechanism, as opposed to a product of high recombination activity among the LTRs followed by adaptive (purifying) selection?

327: “may contribute to trait evolution, thereby facilitating natural selection.” BETTER: may contribute to trait variation, thereby facilitating natural selection.

RESPONSE TO REVIEWERS' COMMENTS

Response:

We would like to thank the reviewers for taking the time to review our manuscript and providing constructive suggestions to improve our work. Below, we provide a point-by-point response (in blue color) to reviewers' comments.

Reviewer #1 (Remarks to the Author)

Guangji Chen et al. "Adaptive expansion of ERVK solo-LTRs in 1 accompanying with Passeriformes speciation"

In this study, the authors investigate the evolution of endogenous retroviruses (ERVs) in birds, and more specifically the solo-LTRs (long terminal repeats) formed by the purging endogenous viral elements (EVEs) from the genome. LTRs make up a very large proportion of the genomes in many animals and plants, but are rarely studied in birds in general. Passerine birds constitute an extraordinary exception from this, and the authors explore the extent and explanation of this based on 362 bird genomes generated from the B10K project. The study is well-designed and very well conducted. The results are highly interesting and discussed in relation to our previous understanding of these processes. I am not an expert on the evolution of ERVs but it seems to me that the referred literature is both appropriate and exhaustive. My comments are restricted to questioning certain claims by the authors, as well as a few typos and minor details.

Response: We appreciate your time and effort on our manuscript. Your constructive and insightful comments are very useful for us to refine our manuscript. We have carefully read your comments and addressed them in this revision.

Line 126: The authors observe a high frequency of solo-LTRs formation in birds, indicating a strong selection against EVEs. They then suggest that this show that birds may have experienced lower rates of paleoviral infiltrations. It is unclear to me why this must be the case. Birds clearly seem efficient in purging EVEs from the genome, but how does this relate to the rate of paleoviral infiltrations?

Response: Previous study by Cui, J. et al., 2014. revealed that birds harbored a limited number of endogenous viral elements compared to mammals and suggested a lower rate of paleoviral infiltrations in birds. We suggest that both the significantly higher level of solo-LTRs formation observed in our data and the lower frequency of paleoviral infiltration suggested by the previous study contribute to the reduced presence of ERVs in avian species. The sentence was modified in the revised manuscript in Line 130-133.

Line 131: Change "neo-ave" to either "neoavian" or "Neoaves".

Response: Done as requested throughout the manuscript.

Line 132: "TE" – is this abbreviation explained somewhere in the text?

Response: Thanks for pointing this out. We have added the abbreviation 'transposable elements (TEs)' as explained in Line 134 to enhance readability.

Line 133 ff.: The authors found a “notably higher” proportion of solo-LTRs in Piciformes and Bucerotiformes but “these were not significantly higher” in other species. To me, the observed non-significance suggests that one cannot say they are higher in these two groups.

Response: We apologize for this confusion. The two comparisons are based on different data, i.e. the inner and outer circle in **Fig. 1a**. The proportions of solo-LTRs relative to genome sizes in *Piciformes* and *Bucerotiformes* were higher than others (shown in the inner circle in **Fig. 1a**), which was consistent with previous studies. But they still have the high frequencies in solo-LTRs formation as other avian species (shown in the outer circle in **Fig. 1a**). To avoid misunderstanding, we have revised this sentence in Line 135-138.

Line 153: avoid using “significant” if not describing the results of a statistical test.

Response: Thanks for your suggestion. We have revised this sentence and revised the ‘significant’ word throughout the manuscript.

Line 155 ff.: The comparisons in the text and in Fig. 2b of the authors present statistics for Passeriformes, Passeri, Passerides and Passerida. The figure 2b shows that there is very little difference between these statistics. This is expected since all lower taxa constitute a subsample of the next higher taxon. As the general point is to show the marked expansion of ERVK solo-LTRs in passerines I think it is enough to discuss and show the statistics for Passeriformes.

Response: Thanks for your comments. We removed other crown groups in **Fig. 2b**, to highlight the ever-accelerating expansion of ERVK solo-LTRs in Passeriformes, especially in the crown group Passerida.

Line 155: The supplementary figure 3 shows that not only ERVK shows the expansion in passerines, this seems true also for ERV1.

Response: Thank you for pointing this out. Indeed, ERV1 solo-LTRs showed the expansion in Passeriformes avian species (**Supplementary Fig. 3**). However, they did not show the ever-accelerating pattern of expansion within the parvorder Passerida, but only expanded once in the ancestral node of Passeriformes. In addition, the proportion of the ERV1 solo-LTR didn't show a significant positive correlation with the number of speciation events within Passerida (**Supplementary Fig. 4b & Supplementary Table 2**). Therefore, we have added a description of the ERV1 expansion in the legend of **Supplementary Fig. 3** and **Supplementary Table 2** to record the results of correlation analyses.

Fig. 2a & part of Supplementary Fig. 3. Proportions of ERV1 and ERVK Solo-LTRs among avian species

Fig 2b-c. ERVK solo-LTRs showed a positive correlation with the number of speciation events in Passeriformes ($r=0.80$, $p\text{-value}=1.61e^{-38}$), especially in the crown group Passerida ($r=0.79$, $p\text{-value}=9.56e^{-10}$).

Supplementary Fig. 4b ERV1 solo-LTRs showed a positive correlation with the number of speciation events in Passeriformes ($r=0.64$, $p\text{-value}=5.71e^{-21}$), BUT not in the crown group Passerida ($r=0.11$, $p\text{-value}=0.5051$).

Line 161: delete “birds”, change “has” to was”.

Response: We have modified it as suggested.

Line 164, supplementary figure 4: The figure shows a positive correlation also for Corvides for ERVL and ERV1.

Response: Sorry for this confusion. We aim to emphasize that for ERVK solo-LTRs, the slope and the correlation coefficient of the regression line is higher in Passerida than in other clades (the slope: 0.0293 for Passerida v.s 0.0073 for Corvides; the correlation coefficient: 0.79 for Passerida v.s 0.37 for Corvides in **Fig. 2c** and **Supplementary Table 2**). But, such a scenario did not exist in ERVL (the slope: 0.0034 for Passerida v.s 0.0060 for Corvides; the correlation coefficient: 0.28 for Passerida v.s 0.26 for Corvides) and ERV1 (the slope: 0.0014 for Passerida v.s 0.0082 for Corvides; the correlation coefficient: 0.11 for Passerida v.s 0.37 for Corvides).

To address these concerns, we added a **Supplementary Table 2** to record the slope values and the results of correlation analyses, and modified the sentence as follows:

- Before: “However, other ERV solo-LTRs did not show this ever-accelerating accumulation pattern (Supplementary Fig. 4).”
- After: “However, other ERV solo-LTRs did not show this ever-accelerating accumulation pattern in Passerida (Supplementary Fig. 4 and Supplementary Table 2).”

Line 164, supplementary figure 4: The x-axis shows number of splits along the phylogenetic lineages. My question is if the number of splits is used as proxy for time, so that the x-axis in realty shows time. Or, if it actually is the number of splits that correlates with the proportion of LTRs, i.e. that the recombination that occurs at these splits somehow is correlated with the proportion of LTRs? Please address this some place in the text or in the figure legend.

Response: We apologize for making you confused. The speciation events on the x-axis of supplementary figure 4 were measured as the number of nodes along a path from the Passeriformes ancestor node to the tips of the B10K family-level bird phylogeny. To clarify this issue, we have added the description in the figure legend as “*Speciation events were measured as the number of nodes along a path from the Passeriformes ancestor node to the tips of the B10K family-level bird phylogeny.*”

Line 166: shouldn't this be “EVRK solo-LTRs”?

Response: Thanks for pointing this out. We have modified it.

Line 167: The p-value for the correlation seems extremely low. The same is true for those in fig. 2b and even more so in suppl. fig. 5. Please check. Also, Pearson's correlation coefficient is “r” (not capital R).

Response: We have modified all the “R” into “r” in **Fig.1b**, **Fig2.b**, and **Supplementary Fig. 5**. In this study, we applied the ‘cor.test’ function from the R package ‘stat’ to calculate the *p-values*. As the reviewer suggested, we examined the calculation process in detail, ruled out the possibility of calculation errors, and provided the raw data needed to reproduce the analyses in **Supplementary Table 3**.

Given that the p-value is a function of both the effect size and the sample size, we ran two simulation schemes with random sampling values of the bivariate Normal (a. same distribution with increasing sample size, b. same sample size with increasing correlation coefficient). The simulation results showed that either the large sample or the intensified correlation could produce extremely low p-values, as what we got.

```
# R
set.seed(123)
m1 <- MASS::mvrnorm(50,mu=c(0,0),Sigma=matrix(c(1,0.6,0.6,1),2))
m2 <- MASS::mvrnorm(100,mu=c(0,0),Sigma=matrix(c(1,0.6,0.6,1),2))
m3 <- MASS::mvrnorm(200,mu=c(0,0),Sigma=matrix(c(1,0.6,0.6,1),2))
dataset_SampleSize<-data.frame(SampleSize=c(50,100,200),
                               pValue=c(cor.test(m1[,1],m1[,2])$p.value,
                                             cor.test(m2[,1],m2[,2])$p.value,
                                             cor.test(m3[,1],m3[,2])$p.value),
                               Cor=c(cor.test(m1[,1],m1[,2])[["estimate"]][["cor"]],
                                     cor.test(m2[,1],m2[,2])[["estimate"]][["cor"]],
                                     cor.test(m3[,1],m3[,2])[["estimate"]][["cor"]]))
dataset_SampleSize
# In same distribution(similar correlation), larger Sample_Size, lower p-value
#   SampleSize      pValue      Cor
# 1         50 2.081131e-06 0.6143305
# 2        100 1.403072e-11 0.6114809
# 3        200 1.283933e-22 0.6198771

m4 <- MASS::mvrnorm(100,mu=c(0,0),Sigma=matrix(c(1,0.3,0.3,1),2))
```

```

m5 <- MASS::mvrnorm(100,mu=c(0,0),Sigma=matrix(c(1,0.4,0.4,1),2))
m6 <- MASS::mvrnorm(100,mu=c(0,0),Sigma=matrix(c(1,0.5,0.5,1),2))
m7 <- MASS::mvrnorm(100,mu=c(0,0),Sigma=matrix(c(1,0.6,0.6,1),2))
m8 <- MASS::mvrnorm(100,mu=c(0,0),Sigma=matrix(c(1,0.7,0.7,1),2))
m9 <- MASS::mvrnorm(100,mu=c(0,0),Sigma=matrix(c(1,0.8,0.8,1),2))
dataset_Cor<-data.frame(SampleSize=c(100,100,100,100,100,100),
                        pValue=c(cor.test(m4[,1],m4[,2])$p.value,
                                cor.test(m5[,1],m5[,2])$p.value,
                                cor.test(m6[,1],m6[,2])$p.value,
                                cor.test(m7[,1],m7[,2])$p.value,
                                cor.test(m8[,1],m8[,2])$p.value,cor.test(m9[,1],m9[,2])$p.value),
                        Cor=c(cor.test(m4[,1],m4[,2])[["estimate"]][["cor"]],
                              cor.test(m5[,1],m5[,2])[["estimate"]][["cor"]],
                              cor.test(m6[,1],m6[,2])[["estimate"]][["cor"]],
                              cor.test(m7[,1],m7[,2])[["estimate"]][["cor"]],
                              cor.test(m8[,1],m8[,2])[["estimate"]][["cor"]],
                              cor.test(m9[,1],m9[,2])[["estimate"]][["cor"]]))

dataset_Cor
# In Sample_Size: larger correlation, lower p-value
#   SampleSize      pValue      Cor
# 1         100 2.968968e-02 0.2175446
# 2         100 3.470760e-07 0.4835250
# 3         100 3.256672e-09 0.5493067
# 4         100 4.335832e-10 0.5738681
# 5         100 7.807956e-17 0.7137340
# 6         100 2.089999e-20 0.7646353

```

Line 203-204: It could well be due to my lack of solid knowledge about this matter, but the sentence that starts with “Such higher...” is unclear to me. Please check if further clarification is needed.

Response: Vertical transmission of new insertions or integrations between generations is only possible through germ cells for ERVs in the host genome Hence, the high transcriptional in the reproductive organs, particularly in the primordial germ cells (Fig. 3b), is the prerequisite for frequent ERVs integration. We have modified it as Reviewer 2 suggested.

Line 213: Fig. 3c: It says the top bars indicate shared LTRs that are higher than two, shouldn't it be “two or higher”

Response: Thanks for your suggestions. We have corrected this.

Line 321: It seems the sentence that starts with “Our study...” should be part of the previous sentence and separated by a comma.

Response: We have revised it as suggested.

Line 334: I suggest replacing “proximal” with “prior”.

Response: We have revised it as suggested.

Line 321: “Our study” should be “our study”.

Response: We have modified it as suggested.

Line 386: Change “1000” to 10,000”.

Response: Thanks for pointing this out. We have corrected these typos.

Line 398: It is unclear to me if the identification of candidate region that is described in this section is done only for the zebra finch or not. Please clarify.

Response: Thank you for bringing this to our attention. This identification pipeline was applied to all species involved in this study, including 362 birds (including the zebra finch), 23 reptiles, and 20 mammals. To make it clearer, we have added a few words to this sentence:

- Before: “*We identified solo-LTRs following the pipeline presented in Supplementary Fig. 1*”.
- After: “*We identified solo-LTRs from the genomes of 405 representative amniote species (including 362 birds, 23 reptiles, and 20 mammals) following the pipeline presented in Supplementary Fig. 1.*”

Supplementary Fig. 7: It seems as the graph is truncated at -2 at the y-axis? If it is I believe it would be better to show the whole figure.

Response: There is no truncation in the graph, and the presentation of the y-axis is due to additional processing for genes of which the expression level is zero or almost zero. For these genes, the logarithm of 0 is represented as minus infinity (-Inf in R). To avoid this issue, we replaced 0 with 0.01 in the gene expression data to ensure that the logarithm yields a finite value (-2 in the graph). We have revised this issue by using $\log_{10}(\text{counts}+1)$ as many previous studies have done (Li and Li 2018, Bossel Ben-Moshe, Hen-Avivi et al. 2019). The **Fig. 3b**, **Supplementary Fig. 7**, and the corresponding *p-values* (from 2.3×10^{-9} into 6.5×10^{-12}) in Line 205 have been updated.

Supplementary Fig. 7: In zebra finches' reproductive organs, ERVK retroviral genes display significantly higher expression levels than other retroviral genes.

We applied a $\log_{10}(\text{count} + 1)$ transformation to the normalized count for visualization.

Reviewer #2 (Remarks to the Author)

The authors present an extensive analysis of the evolution of LTR retrotransposons throughout the avian phylogeny using the large genomic dataset provided by the B10K project. The authors investigate the presence of species-specific LTRs, their polymorphisms and their diversity in the different species. Thanks to the use of transcriptomic data on top, the authors are able to measure the transcription of LTRs and genes in several tissues including primordial germ cells. They were able to identify some candidate LTRs that may serve as gene promoters and test the functionality of one of them in vitro finding positive evidence for its function as promoter.

I find the analysis very interesting and relevant to the field, but I have some minor and major points to address.

Response: We appreciate the reviewer's interest in our work. We have revised the manuscript accordingly according to the constructive and insightful comments of the reviewer.

Minor

1) Line 49-52: I would rephrase more generally in “which results in the deletion of the internal region” so to include all those instances of truncated, non -autonomous ERVs that lost any viral ORF.

Response: We have reworded it as you suggested.

2) Line 72: I would rephrase in “the widely studied syncytin1 gene, derived from the HERV-W env”.

Response: We agree with the reviewer's comment. We have modified it as you suggested.

3) Line 93-96: This is a strong statement since more neutral processes can lead to the same pattern, like the higher frequency of recombination in avian genomes compared to mammals. On top of that, the rates of homologous recombination and non-allelic homologous recombination - responsible for the formation of solo LTRs - are associated as they are part of the same molecular phenomenon. Selection has not been tested in any on the analysis therefore I would highly recommend the authors to stick to more neutral explanations and add selection only as a speculation.

Response: Thanks for this constructive comment. We agree with the reviewer and revised relevant sentences accordingly.

Revised Manuscript Line 129:

- Before: “*indicating a strong selection force to maintain a low proportion of EVEs in the genome.*”
- After: “*indicating a selection force might have functioned to maintain a low proportion of EVEs in the genome.*”

Revised Manuscript Line 346-350:

- Before: “*In contrast, throughout the evolution of birds, strong selection pressures to purge ERVs have resulted in the intervening proviral sequences for most avian*

ERVks being eliminated through a substantially higher frequency of ERVK solo-LTR formation.”

- After: “*In contrast, throughout the evolution of birds, both potential selection pressures to purge ERVs and high recombination activity among the LTR might have resulted in the intervening proviral sequences for most avian ERVKs being eliminated through a substantially higher frequency of ERVK solo-LTR formation.”*

4) Line 188-189: I find the phrasing here very confusing. Passerida still have around 10% of genome repetitive, how is it possible that solo LTRs make up 76% of the genome?

Response: We apologize for this confusion. 76% is the ratio of solo-LTRs to total LTRs length across the genome. We used this index to represent the frequency of solo-LTRs formation. We have revised this sentence.

- Before: “*We also found that ERVK solo-LTRs shared between any two Passeri species under the parvorder Passerida made up a large proportion (an average of 76.69%) of their genomes.”*
- After: “*We also found that ERVK solo-LTRs shared between any two Passeri species under the parvorder Passerida made up a large proportion (an average of 76.69%) of all ERVK solo-LTRs in their genomes.”*

5) Line 192 and throughout the manuscript: the abbreviations should be introduced in parentheses after their complete form outside the parentheses.

Response: We appreciate your keen observation. We have modified these throughout the manuscript.

6) Line 203-205: the reproductive organs are very heterogeneous tissues therefore expression in the whole organs is not necessarily a prerequisite for integration in the next generation. Activity in the primordial germ cells can be. Please rephrase.

Response: Thanks for your suggestion. We have revised this sentence as suggested.

7) Figure 3: I would not say that the transcription of ERVs is evidence for their ongoing accumulation, it is a necessity but not sufficient for their transposition and as evidence for their ongoing integration, therefore please change the caption of the figure.

Response: We agree with the reviewer's comment on the caption of **Figure 3**. In the figure, we showed the transcription of ERVs (panel b) and the signal of recent population diversity in ERVK solo-LTRs (panel c). These patterns are more indicative than definitive evidence. We have replaced 'Evidence' with 'Indication'.

Major

1) Line 230-231: 10 kb is a rather large threshold, what does it happen with smaller thresholds? These thresholds are completely arbitrary so it would be good to explore the effects of different thresholds on your results.

Response: Thanks for your suggestion. Three thresholds of 2 kb, 5 kb, and 10 kb were used for GO enrichment analysis, respectively. Across all three criteria, a total of 22 GO terms were found to be consistent. The detailed results of the enrichment analysis have been added into **Supplementary Table 6**. In the main text, GO enrichment analysis with a 10 kb threshold was provided, and we believe that this threshold did not overestimate the distance between the regulatory region and the adjacent genes. In a study published by (Pan, Wang et al. 2023), the authors stated that the median distance of gene-regulatory elements was 249,346 bp in the chicken. We would clarify that using 10 kb to present the results is a relatively reasonable criterion.

2) I find the description of the method to identify solo LTRs is not clear enough in its steps even in the supplementary, so I kindly ask the authors to make it clearer as it is the foundation for the entire manuscript.

Response: Thanks for highlighting this issue. We have improved the method for identifying solo-LTRs, refined the description of the details in the methods section (Line 400-429), and provided the **Supplementary Note** to show the results of the evolutionary patterns of solo-LTRs with the target site duplications following the comment #3.

3) As far as I understand, I do not really see in the method a step in which the authors distinguish solo LTRs from fragmented LTR retroelements. I think full-length elements are masked away from potential solo LTRs in the first step of the pipeline with LTRharvest but I do not think this is sufficient for a couple of reasons: 1) the genomes used from the B10K project are highly fragmented which hinders drastically the recovery of full-length elements because not assembled at all thus biasing the counts; 2) fragmented LTR retroelements that can result from independent deletion events and/or internally truncated elements will not be clearly identified by LTRharvest since they lost the protein coding genes. Both points will inflate the number of solo LTRs in comparison to full-length and inflate the number of false positives. I suggest following methods in which target site duplications at the extremities of the potential solo LTR are taken into consideration like in <https://doi.org/10.1101/2021.12.31.473444>

Response: Thanks for your suggestion. Considering the removal process of transposons would lead to the target site duplications (TSDs) being imperfect and generating sequence diversity (Fedoroff 2012), it might cause the specific boundaries of TSDs and LTRs to not be so clear (Ji and DeWoody 2016). To address this, we not only checked the target site duplications (TSDs) in the proximal 10 bp upstream and downstream for the paired-LTRs (intact LTRs) and solo-LTRs as employed in (Peona, Blom et al. 2022), but also extended the flanking region into 15 bp and 20 bp. We employed blast with the criterion 4 to 6 bp complete hits by following the method from (Peona, Blom et al. 2022). All major conclusions and findings were consistent with our previous results and were consistent across the three criteria (details were provided in the **Supplementary Note: Evolutionary patterns of solo-LTRs with the target site duplications**). We agree with the reviewer that, using TSD as a standard to identify intact LTRs and solo-LTRs is conducive to reduce the false positive of solo-LTRs identification and make our conclusions more solid.

Detailed results were as follows:

Supplementary Fig. 11: Phylogenetic tree with the proportion of solo-LTRs (the inner circle) and the ratio of solo-LTR (outer circle) in birds, reptiles, and mammals under different TSDs criteria.

Supplementary Fig. 12: Solo-LTRs counts were positively correlated with the genome size in mammals and reptiles, but not in birds under different TSDs criteria.

Supplementary Fig. 13: ERVK solo-LTRs accumulated in Passeriformes under different TSDs criteria.

Supplementary Fig. 14: ERVK solo-LTRs accumulate during speciation events in Passeriformes, especially in Passerida, under different TSDs criteria.

Supplementary Fig. 15: Shared ERVK solo-LTRs constitute a large portion of all ERVK solo-LTRs in Passerida bird species under different TSDs Criteria.

Reviewer #3 (Remarks to the Author):

Review of Chen et al., “Adaptive expansion of ERVK solo-LTRs in accompanying with Passeriformes speciation” for Nature Communications

The paper by Chen et al. is an interesting application of large-scale comparative genomics to assess potential mechanisms shaping evolution. In this case, the mechanism of interest is the generation of ERV solo-LTRs, with their potential to regulate the expression of adjacent genes.

Here, they first show that Solo-LTR numbers are positively correlated with genome size in mammals and reptiles, but not in birds. They interpret this as evidence for “strong selective pressures to purge ERVs from bird genomes.” This is not exactly a novel observation (since we already knew that birds have smaller genome sizes and reduced repetitive sequence content), but it does advance the field by highlighting a specific mechanism that may be especially active in birds for pushing back against transposon-mediate genome size expansion. (But why this is so in birds remains a mystery!).

Next, they show that ERVK solo-LTRs, in particular, appear to increase during more recent speciation events, especially within Passeriformes. Indeed, almost half of ERVK solo-LTRs were species-specific in the suborder Passeri and appear to have emerged through “multiple recent bursts in a lineage-specific way.” Moreover, using whole genome re-sequencing data from 19 zebra finch individuals, they found that 655 (2.68%) ERVK solo-LTRs were polymorphic among individuals, suggesting an active reservoir of recent (and probably ongoing) genetic variation. This is not a radical discovery in general (retrotransposons are active!), but it’s nice to see this direct and detailed characterization in songbird lineages and the zebra finch.

To assess the potential functional impact of ERVK insertions, they mined public RNAseq data for the zebra finch, finding evidence for transcriptional activity among 120 ERVK genes. They looked at 68 genes that harbored new ERVK solo-LTs in zebra finch relative to chicken and found that 20 were differentially expressed between the two species. They then focused on the ITAG2 gene, building on prior evidence that its expression is correlated with song rate in another songbird species (white-throated sparrow). Looking in the zebra finch, they find an ERVK solo-LTR in the ITAG2 upstream region overlapping with an H3K27ac ChIP-Seq signal. From this, they developed a plausible (but still “just so”) story suggesting how this flanking ERVK solo-LTR might support the evolution of singing behavior. Indirectly supporting this, they transfected this element into chicken fibroblasts and found evidence that it can indeed increase the expression of a flanking reporter gene. I don’t see any major flaws in the analysis and presentation, though I would direct the authors to the following specific details and considerations.

Response: We appreciate your professional review work on our manuscript. We have carefully read your comments and addressed them in this revision.

Line 37 and elsewhere: HVC is referred to as an acronym for “High Vocal Center.” To my knowledge, no one has defined HVC as an acronym since the nomenclature was revised in 2004 (Reiner et al., “Songbirds and the Revised Avian Brain Nomenclature”. *Annals of the New York Academy of Sciences*. 1016 (1): 77–108. doi:10.1196/annals.1298.013). Rather,

the community seems to favor “HVC (used as a proper name)” and then citing the Reiner paper.

Response: Thank you for your helpful suggestion. We have modified the words into “*HVC (used as a proper name)*” throughout the manuscript and cited the literature of Reiner et al.

Line 145: “Bird species with potentially problematic assemblies (genome size < 800 Mb or scaffold N50 < 20 kb) were filtered to reduce the bias of assembly quality.” I assume they mean they removed these assemblies from the analysis? But this actually raises another consideration: Might there be some systematic difference in the way the Bird 10K assemblies were assembled with respect to repetitive sequence content compared to the non-avian assemblies? For example, how many assemblies were based on short-read vs long-read sequencing technologies? Is there a reason we can discount this as a possible confounder in the analysis?

Response: Thank you for pointing this out. Your concerns did align with ours. We removed these poor assemblies to avoid the correlation bias associated with the assembly's quality.

In our study, the genomes of birds are mainly assembled at the scaffold level, rather than at the chromosome level in non-avian genomes. To reduce false positives for solo-LTRs identification, we not only filtered out low-quality avian genomes but also discarded excessively short scaffolds (less than 20k) in the statistics. Differences in the evolution patterns of solo-LTRs between avian and non-avian species remained stable under multiple filtering criteria, as shown in **Supplementary Fig. 10** and **Supplementary Table 7**. Further, following Reviewer #2's comment, we examined the target site duplications (TSDs) at the extremities of the potential solo-LTRs, to distinguish the solo-LTRs from the fragmented LTR retroelements and to obtain the high-quality solo-LTRs for downstream analysis. Overall, the evolutionary patterns of solo-LTRs were consistent, using either all potential solo-LTRs or the solo-LTRs with TSDs (**Supplementary Fig. 11-15**).

In addition, we conducted an additional examination of the impact of assembly quality on the solo-LTRs identification within avian taxa (**Response Letter Fig. 1**). At both the solo-LTR counts and frequency solo-LTR formation levels, correlations exhibited the r-value less than 0.19 (0.17 for counts and 0.12 for frequency). According to (Zhu 2012, Zhu 2016), a correlation r-value below 0.19 is considered to indicate 'No correlation'. In light of these potential influences, we have incorporated these limitations into the discussion section, and emphasized the need for further exploration across a broader taxonomic range with high-quality genomes in Line 361-363.

Response Letter Fig. 1: Solo-LTRs showed the non-correlation ($r < 0.19$) with Scaffold N50 on the (A) Counts level and (B) Ratio of formation.

Line 161 (missing word): “The speciation process of Passeriformes birds has BEEN accompanied by an accumulation of ERVK solo-LTRs.”

Response: We have corrected this.

Lines 240-245: if you just picked any 68 genes at random, what proportion would be differentially expressed between ZF and chicken? (Knowing this would confirm the statistical significance of their observation that about a third of ERVK-associated genes are differentially expressed).

Response: Thank you for pointing this out. These 68 ERVK-associated genes harbored the insertion of ERVK solo-LTRs in zebra finch specifically, which contributed potential new regulatory regions. 20 of 68 genes (29.4%) were significantly differentially expressed between the chicken and zebra finch in the brain.

Out of a total of 332 orthologous genes harboring the ERVK solo-LTRs in the 10kb flanking regions both in zebra finch and chicken, 130 exhibited significant differential expression, with 100 up-regulated in zebra finch and 30 down-regulated. Among any randomly selected set of 68 genes, the proportion of differentially expressed genes was higher than one third (around 39.2%).

Additionally, it's essential to acknowledge that these differential expression patterns are estimated from the brain transcriptomic data. The significant observation expected by the reviewer was not present on the brain tissue.

Line 256: is there evidence for the flanking ERVK solo-LTR in the genome of the white-throated sparrow (*Zonotrichia albicollis*)?

Response: Based on the available data, there is no evidence to show that the *ITGA2* of white-throated sparrow (*Zonotrichia albicollis*) is flanked by ERVK solo-LTRs. The nearest ERVK solo-LTR is 284kb apart from the *ITGA2* of *Zonotrichia albicollis* in GCA_000385455.1. The 405bp ERVK solo-LTRs of the zebra finch (*Taeniopygia guttata*) could only be traced to

the common ancestor of the white-rumped munia (*Lonchura striata*) and zebra finch (*Lonchura striata* is the most closely related species of zebra finch in this study).

Lines 320-321: an incomplete sentence.

Response: We have corrected this.

Line 324: “The formation of solo-LTRs may serve as a host defense mechanism for purging newly inserted ERVK” -- this implies a discrete mechanism focused on defense. That would be interesting if true. But is there any support for this as a defense mechanism, as opposed to a product of high recombination activity among the LTRs followed by adaptive (purifying) selection?

Response: Thanks for raising this insight question. We agree it is difficult to distinguish between the discrete defense mechanism and the high recombination activity followed by adaptive selection. To avoid any misunderstanding, we have rewritten the sentence as follows:

- Before: “*We propose that in Passerida species, the formation of solo-LTRs may serve as a host defense mechanism for purging newly inserted ERVK, thereby counteracting the deleterious consequences of the expansion of ERVK in this group.*”
- After: “*We propose that in Passerida species, the formation of solo-LTRs may serve as a host defense mechanism for purging newly inserted ERVK or as a consequence of the high recombination activity among the LTRs, thereby counteracting the deleterious consequences of the expansion of ERVK in this group.*”.

Line 327: “may contribute to trait evolution, thereby facilitating natural selection.” BETTER: may contribute to trait variation, thereby facilitating natural selection.

Response: Suggestion is accepted.

Reference:

1. Bossel Ben-Moshe, N., S. Hen-Avivi, N. Levitin, D. Yehezkel, M. Oosting, L. A. B. Joosten, M. G. Netea and R. Avraham (2019). "Predicting bacterial infection outcomes using single cell RNA-sequencing analysis of human immune cells." *Nature Communications* 10(1): 3266.
2. Fedoroff, N. V. (2012). "Transposable Elements, Epigenetics, and Genome Evolution." *Science* 338(6108): 758-767.
3. Ji, Y. and J. A. DeWoody (2016). "Genomic Landscape of Long Terminal Repeat Retrotransposons (LTR-RTs) and Solo LTRs as Shaped by Ectopic Recombination in Chicken and Zebra Finch." *Journal of Molecular Evolution* 82(6): 251-263.
4. Li, W. V. and J. J. Li (2018). "An accurate and robust imputation method scImpute for single-cell RNA-seq data." *Nature Communications* 9(1): 997.
5. Pan, Z., Y. Wang, M. Wang, Y. Wang, X. Zhu, S. Gu, C. Zhong, L. An, M. Shan, J. Damas, M. M. Halstead, D. Guan, N. Trakooljul, K. Wimmers, Y. Bi, S. Wu, M. E. Delany, X. Bai, H. H. Cheng, C. Sun, N. Yang, X. Hu, H. A. Lewin, L. Fang and H. Zhou (2023). "An atlas of regulatory elements in chicken: A resource for chicken genetics and genomics." *Science Advances* 9(18): eade1204.
6. Peona, V., M. P. K. Blom, C. Frankl-Vilches, B. Milá, H. Ashari, C. Thébaud, B. W. Benz, L. Christidis, M. Gahr, M. Irestedt and A. Suh (2022). "The hidden structural variability in avian genomes." *bioRxiv*: 2021.2012.2031.473444.
7. Zhu, W. (2012). "Sadly, the earth is still round ($p < 0.05$)." *Journal of Sport and Health Science* 1(1): 9-11.
8. Zhu, W. (2016). " $p < 0.05$, < 0.01 , < 0.001 , < 0.0001 , < 0.00001 , < 0.000001 , or < 0.0000001 " *Journal of Sport and Health Science* 5(1): 77-79.

REVIEWERS' COMMENTS

Reviewer #1 (Remarks to the Author):

This is my second time reviewing this manuscript. My previous concerns have been satisfactorily addressed, and I find the manuscript suitable for publication.

Reviewer #2 (Remarks to the Author):

I think the authors addressed my comments in a detailed way. I do not have further comments to make.

Reviewer #3 (Remarks to the Author):

The authors have been very responsive to the previous review. I had already found the paper to be solid, but had a few questions which they have considered in their rebuttal, and I am satisfied that the work is ready to go forward. My only concern now is that I find the title a bit grammatically awkward, "in accompanying with". Maybe "in association with"?

RESPONSE TO REVIEWERS' COMMENTS

Response:

We would like to thank the reviewers for taking the time to review our manuscript and agreeing to publish our work.

Reviewer #1 (Remarks to the Author):

This is my second time reviewing this manuscript. My previous concerns have been satisfactorily addressed, and I find the manuscript suitable for publication.

Response:

Thanks for your constructive suggestion and recognize our work.

Reviewer #2 (Remarks to the Author):

I think the authors addressed my comments in a detailed way. I do not have further comments to make.

Response:

Thanks for your constructive suggestion to improve our work for publish.

Reviewer #3 (Remarks to the Author):

The authors have been very responsive to the previous review. I had already found the paper to be solid, but had a few questions which they have considered in their rebuttal, and I am satisfied that the work is ready to go forward. My only concern now is that I find the title a bit grammatically awkward, "in accompanying with". Maybe "in association with"?

Response:

Thanks for your suggestion. After fully considering your and the editorial team's suggestions for the title, we have revised the title into: "Adaptive expansion of ERVK solo-LTRs is associated with Passeriformes speciation events".